

# Low-resolution Australasian palaeoclimate records of the last 2000 years

Bronwyn C. Dixon[1], Jonathan J. Tyler[2], Andrew M. Lorrey[3], Ian D. Goodwin[4], Joëlle Gergis[5], Russell N. Drysdale[1]

[1]School of Geography, University of Melbourne, Melbourne, 3010, Australia
[2]Department of Earth Sciences, University of Adelaide, Adelaide, 5005, Australia
[3]National Institute of Water and Atmospheric Research, Auckland, 1010, New Zealand
[4]Marine Climate Risk Group and Department of Environmental Sciences, Macquarie University, Sydney, 2109, Australia
[5]School of Earth Sciences, University of Melbourne, Melbourne, 3010, Australia

*Correspondence to*: Bronwyn C. Dixon (bdixon1@student.unimelb.edu.au)

**Abstract**

Non-annually resolved palaeoclimate records in the Australasian region were compiled to facilitate investigations of decadal to centennial climate variability over the past 2000 years. A total of 661 lake/wetland, geomorphic, marine, and speleothem records were identified and then assessed against a set of *a priori* criteria based on temporal resolution, record length, dating

methods, and confidence of the proxy-climate relationship over the Common Era. A high quality subset of 22 records across Australasia met the criteria and they are endorsed for subsequent analyses. New chronologies based on progressive Bayesian techniques were constructed for the high quality records to ensure a consistent approach to age modelling and quantification of age uncertainties. Chronological uncertainty was the primary reason why records did not meet the selection criteria. Despite present limitations, existing proxies and reconstruction techniques that successfully capture climate variability in the

region show potential to address spatial gaps and expand the range of climate variables covering the last 2000 years from the Australasian region.

## 1 Introduction

### 1.1 Background

Elucidation of climate system variability and long-term change, especially for investigation of anthropogenic impacts,

requires a spatially and temporally robust baseline of natural climate conditions. This is particularly relevant for Australasia, where climate variability has strong agricultural, socioeconomic, and environmental impacts (Head et al., 2014, Kiem et al., 2016, van Dijk et al., 2013). The Intergovernmental Panel on Climate Change (IPCC) previously recommended a two thousand year baseline to establish climate variability for Earth (Masson-Delmotte et al., 2013). This time period is ideal for providing an effective context for three primary reasons: (i) shifts in global boundary conditions are believed to be minor and

well-constrained over this time frame; (ii) the last two millennia contains periods of relatively warm and cool climate



anomalies, which offers opportunity to study regional responses to global and/or regional climate fluctuations; and (iii) the majority of high-resolution proxy climate records occur within this period, which provides reconstructions of climate at individual sites and across geographic regions with increased chronological certainty (PAGES2k Consortium, 2013, Neukom and Gergis, 2012).

The International Geosphere-Biosphere Programme-led Past Global Changes (PAGES) 'Regional 2k' initiative was established in 2008 to coordinate and synthesise regional and continental-scale regionstructions spanning the last 2000 years. This network also provides a way to exchange ideas and test hypotheses about recent palaeoclimates that can be queried using consistent pre-screened datasets. The PAGES initiative encourages use of rigorous, multi-method approaches to compare variability, patterns, and timing of changes across regions (Kaufman, 2014). The 'Aus2k' working group was made

responsible for investigation of the Australasian region climate for the last 2000 years, hereafter referred to as the Common Era (CE).

       Data derived from natural archives for the purpose of investigating climate variability falls into two classifications: 'high-resolution' (monthly to annually resolved) and 'low-resolution' (non-annually resolved). High-resolution archives, such as tree-rings, corals, varved sediments, and some ice cores, have been used to reconstruct regional and global

temperature trends over the Common Era (PAGES2k Consortium, 2013, Gergis et al., 2016, Mann, 2007, Mann and Jones, 2003). In addition to several advantages of using annually resolved data, such as chronological precision and temporal resolution, annually resolved data possess a number of limitations when examining the past two millennia. These limitations can include the confined spatial coverage inherent to the high-resolution data network, and difficulties in capturing low-frequency variability (Anchukaitis and Tierney, 2012). The limitations of interpreting low-frequency signals in high-

resolution archives are exemplified with tree rings, where multiple short, individual series are used to compile longer composite chronologies. The applied standardisation techniques used to construct the composite chronologies can restrict meaningful long-term signal retention, referred to as the 'segment length curse' (Cook et al., 1995).

       Due to the limited potential of reconstructing multi-decadal and longer-term variability from high-resolution archives, it is vital to complement the resulting reconstructions with non-annually resolved, low-resolution records (Moberg

et al., 2005). Previous studies have explored the blending of high- and low-resolution archives in the Australasian region to investigate a range of climate variables and atmospheric circulation patterns (Goodwin et al., 2013, Goodwin et al., 2014, Lorrey et al., 2007, Lorrey et al., 2008). Annually resolved records in the Southern Hemisphere have been identified, reviewed, vetted, and augmented (Neukom and Gergis, 2012), but no equivalent effort currently exists for non-annually resolved records in the Australasian region.

Low-resolution sedimentary archived available within Australasia include lacustrine, fluvial and wetland sediments, marine sediments, speleothems, and geomorphic features (e.g. moraines, dunes) across diverse climate settings. These archives have multiple strengths for use in palaeoclimate reconstructions. First, the typical length of low-resolution records (>500 years) is longer than many available annually resolved records. Data processing techniques for low-resolution sedimentary archives also tend to preserve low-frequency climate signals, i.e. those over multi-centennial and longer time



frames. Second, low-resolution archives have a greater potential to be more widely distributed when compared to annually resolved archives for most regions (Anchukaitis and Tierney, 2012). This is especially true in the ocean-dominated Southern Hemisphere. For example, marine sediment cores can provide records of oceanic variability without the tight climatic and bathymetric constraints of coral records. Other low-resolution archives such as speleothems and lake sediments can provide

continuous records from climate regions spanning arid regions (Gliganic et al., 2014, Quigley et al., 2010) to the tropics (Crausbay et al., 2006, Rodysill et al., 2013). Third, the climate signals preserved in low-resolution records can differ from those found in annually-resolved records, meaning that a wider range of long-term climate changes, as well as differing types of climate variability, can be reconstructed through their inclusion in multi-proxy syntheses.

**1.2 Aims**

To date, the Australasian region has hundreds of published low-resolution palaeoclimate records. However, a comprehensive palaeoclimate record database has not been compiled and evaluated for suitability to contribute to examination of climate variability during the Common Era. In this study, all low-resolution palaeoclimate datasets currently available for Australasia were compiled and assessed for suitability for quantitative climate reconstruction efforts. Records were assessed against a set of pre-determined quality metrics to develop a subset of high-quality records that could then be presented in a

flexible, consistent format to facilitate future palaeoclimate investigations in the region. The four objectives of this study are to: (i) identify Australasian palaeoclimate records spanning the last 2000 years; (ii) evaluate each record against the criteria set out by the global PAGES2k network, (iii) recalibrate age-depth models of the resulting 'high-quality' subset of records using Bayesian techniques; and (iv) discuss merits and shortcomings of existing palaeoclimate reconstructions while providing recommendations for future palaeoclimate record collection and development for the Australasian scientific

community.

       The work compliments a recent review of all annually resolved records in the Southern Hemisphere covering the Common Era (Neukom and Gergis, 2012), as well as other sub-region synthesis efforts in Australasia (Freeman et al., 2011, Gouramanis et al., 2013, Lorrey et al., 2007, Lorrey et al., 2008, Lorrey et al., 2010). The outcomes of this study will be helpful to researchers by identifying the most reliable low-resolution records in the Australasian region for climate model-

palaeoclimate data intercomparisons in the Southern Hemisphere (Phipps et al., 2013).

**2 Data and Methods**

Published records were identified through inspection of citation databases, reference lists of past review papers, and online public data repositories. Data were retrieved from both the National Oceanic and Atmospheric Administration (NOAA) paleoclimate database (http://www.ncdc.noaa.gov/data-access/paleoclimatology-data) and the Neotoma database

(http://www.neotomadb.org).



### 2.1 Defining Aus2k low-resolution palaeoclimate data criteria

For this study, all assembled records were screened against a set of selection criteria for the purpose of identifying a 'high-quality' subset for the use of the PAGES Aus2k working group. The selection criteria, outlined below, follow those designated by the international PAGES Regional 2k initiative (http://pastglobalchanges.org/ini/wg/2k-network/data).  For

inclusion, a record was required to meet the following five criteria:

(i) The proxy must be related to one or more climate variables, as stated in a peer-reviewed publication.

(ii) The record must extend continuously for at least 500 years out of the last 2000 years.

(iii) The record must have an age model based on at least two to three chronological anchors.

(iv) The record must have an average sample resolution between 2-50 years per sample or analyses.

(v) The collection location must fall within the region that has been identified by PAGES Aus2k to influence Australasian climate (90°E - 140°W, 10°N - 80°S) (Gergis et al., 2016). The Australasian region includes tropical Southeast Asia because of the dynamical influences of the Indo-Pacific region on the Australasian monsoon.

The application of each criterion was justified on several grounds. First, records must have a published, peer-reviewed interpretation of the relationship between the measured proxy and climate variable(s). This provides quality control through

the traditional peer review process by palaeoclimate specialists. Second, the overall length and average sample resolution have been selected for the purpose of reconstructing climate variability at a range of time scales over the Common Era. It is important that the data have sufficiently high resolution to capture fluctuations at multi-decadal scales, and that they are long enough to capture an appropriate number of cycles in the study period (Chen and Grasby, 2009). Third, the low-resolution records must have reasonable chronological constraints. 'Reasonable' was defined as PAGES2k as containing at least one

chronological control point near the youngest part of the record, another near 1CE or the end of the record (whichever is younger), and, for records greater than 1000 years in length, an additional date near the middle of the record. Many records, primarily in the marine realm, extend beyond the Common Era. Dates outside this period were retained to constrain interpolation uncertainties at age-model extremes. The final requirement is that records must be publically available, which contributes towards the transparency and reproducibility of results. All of the records identified within the Aus2k region

have been previously published, but very few were publically available. One of the tasks undertaken by the Aus2k working group was to ensure that all records were stored in public archives. All Aus2k datasets will be archived through their use in PAGES2k global syntheses.

Preservation of metadata relating to the Aus2k datasets will assist future computer-driven, multi-record comparisons and reconstructions. One recent initiative of the palaeoclimate community is the development of a 'common

language' of palaeoclimatology, where data and metadata are stored in a consistent, machine-readable format (Emile-Geay and Eshleman, 2013, McKay and Emile-Geay, 2015). The Linked Paleo Data (LiPD) framework has been developed with input from the wider scientific community (McKay and Emile-Geay, 2015). The metadata retained for the Aus2k datasets are based on the fields in the LiPD framework. As a part of the metadata, raw age determinations are recorded. Although one



approach for the creation of age models in presented in this study, the raw data are available for individuals who wish to apply alternative methods.

### 2.2 Past palaeoclimate syntheses in Australasia

Previous palaeoclimate and palaeoenvironmental reviews have compiled lists of studies and obtained proxy data to address a range of research questions from the Australian and New Zealand sectors. Australian palaeoclimate records containing at least one date in the Holocene were compiled through a partnership between Macquarie University and the New South Wales Department of Environment, Climate Change, and Water (Freeman et al., 2011). The review identified 190 low-resolution records and 51 high-resolution records. Or these 241 records, 141 were classified as 'moderate to high confidence' based on climatic sensitivity, possible non-climatic influences, local forcing, and chronological confidence. Gouramanis et al. (2013) compiled a set of records for a latitudinal transect across the Australian continent, used to compare consistency in climate histories between eastern and western Australia over the past ~8000 years. The datasets were chosen by the reputation of the published records, based on the regularity of citation.

Other groups within Australasia have previously conducted multi-proxy palaeoclimate syntheses under the banner of AUS-INTIMATE (Australasian INTegration of Ice, MArine, and TErrestrial records). Records identified through this initiative have benefitted the most recent PAGES2k effort to evaluate and update the last 2000 years. AUS-INTIMATE was organised within the International Union for Quaternary Research (INQUA), with two distinct research phases from 2004-2007 and 2008-2011. Both NZ- (New Zealand) and OZ- (Australia) INTIMATE projects focused on the interval 30,000-8,000 years ago (Alloway et al., 2007, Barrell et al., 2013, Bostock et al., 2013, Reeves et al., 2013a); however, many of the compiled records extend to the present. The purpose of AUS-INTIMATE was to integrate available proxy data into climate event stratigraphies. AUS-INTIMATE coverage has been extended from 8000 years ago to present via the Southern Hemisphere Assessment of PalaeoEnvironments (SHAPE) project supported by INQUA since 2013.

New Zealand palaeoclimate proxies covering several late Quaternary intervals, including the last 2000 years, were compiled prior to and during Phase I of Aus2k (Lorrey et al., 2008, Lorrey et al., 2010) to address how regional atmospheric circulation has changed during the late Holocene. The New Zealand data assemblage for the last 2000 years included 35 low-resolution (multi-decadal to centennial) hydroclimatically sensitive records. These data were compiled from a site level up to a homogeneous regional climate district level (Kidson, 2000, Lorrey et al., 2007) from a range of environments that range from temperature subtropical in the far north of New Zealand to glacial in the south. Thirteen temperature-sensitive records were obtained from studies that had undertaken multi-decadal binning of annually resolved temperature reconstructions. There are also several speleothem stable isotope series for New Zealand that cover the Holocene, including the last 2000 years (Lorrey et al., 2008, Lorrey and Bostock, 2017, Lorrey and et al., 2010). In addition, two compilations of past Southern Alps glacier activity (summarised from Lorrey et al., 2008 and Schaefer et al., 2009) and a summer mean surface pressure anomaly reconstruction (after Villalba et al., 1997) completed the New Zealand low-resolution dataset that is relevant for



Aus2k. For the purpose of not repeating data included in the Aus2k high-resolution compilation, the New Zealand tree ring data are excluded here.

Additional global syntheses are also drawn upon in this study. The charcoal database presented by Mooney et al. (2011) is the Australasian subset of a global charcoal database (Power et al., 2010), and contains all datasets (published and

unpublished) up until 2011 that include charcoal analyses. The OZ-PACS working group under the Australasian Quaternary Association (AQUA) was created to investigate ecosystem changes and human impact on the landscapes over the last 500 years. One outcome of the initiative was a list of records until 2007 that covered the target period (Fitzsimmons et al., 2007).

**2.3 Caveats on resolving changes from low-resolution sedimentary records**

Despite the benefits of low-resolution archives, these records often contain relatively large errors associated with radiometric

dating techniques (Anchukaitis and Tierney, 2012, Moberg et al., 2005). Age ambiguity leads to difficulty in cross-correlating records. These errors can result from three primary complications in radiocarbon dating techniques: dead carbon contamination, multiple possible calibrated ages associated with measured $^{14}$C values, and inbuilt ages (McFadgen, 2007).

Radiometrically 'old' or 'dead' carbon derived from groundwater and/or carbonate rocks can drastically alter the apparent $^{14}$C age of freshwater bodies (Geyh, 2000) and the organisms that live within or near the water (Beavan-Athfield

and Sparks, 2001). The possibility of contamination by old carbon can be minimised through the use of short-lived plant macrofossils (Blois et al., 2011), when present, as well as dating of modern water samples to identify the presence of an old carbon reservoir (e.g. Gouramanis et al., 2010). Fluctuations in atmospheric $^{14}$C through time may result from aperiodic changes in ocean-atmosphere-terrestrial radiocarbon partitioning, upper atmosphere radiocarbon production, and oceanic upwelling dynamics (Rodgers et al., 2011) that lead to 'wiggles' in the radiocarbon calibration curve. When this is pared

with the measurement uncertainty of radiocarbon samples, it leads to a range of possible calendar dates (Blaauw, 2010). The inbuilt age of samples may have multiple components that influence dating uncertainties. These include incorporated age (i.e. chronologically old material contained in an organism that pre-dates its death, as with long-lived trees) and environmental residence time (also termed 'storage age'), which is the elapsed time between the death of the dated organism and deposition of its remains in a dated horizon (McFadgen, 1982). The most common example of this phenomenon is the

preservation of long-lived trees in the landscape before deposition in sediment (see Grant, 1985, for an example related to alluvial sedimentation).

It is not always possible to identify environmental residence time within a radiocarbon samples, and the uncertainty may not be acknowledged within the resulting chronology (McFadgen, 2007). In some cases, anatomical studies may reveal whether incorporated age is large or minimal (e.g. identifying heard wood versus sapwood in tree samples). Methods exist to

attempt to classify the relationship of a radiometric age to a horizon of interest, for example: i. a *close age* is a radiocarbon date from within or directly adjacent to the horizon of interest, which is assumed to have a minimal inbuilt age, due to the type of material; ii. a *minimum age* is derived from a sample with a minimal inbuilt age, which has been collected stratigraphically above the horizon or event of interest; and iii. a *maximum age* is derived from a sample with minimal inbuilt





age, which has been collected from a horizon below the horizon or event of interest (McFadgen, 2007). Overall, the use of close ages within a horizon of interest, in conjunction with bracketing by minimum and maximum ages, assists in decreasing uncertainty in dating individual events. Estimated inbuilt ages for some common New Zealand plant species have been published (McFadgen, 2007), but less emphasis has been placed on this subject in Australian studies.

5         Identifying a high-quality subset of records allows both an inter-comparison of records and helps to answer questions of climatic consistency between or within regions (Tyler et al., 2015). Two factors that complicate comparison of published studies are the reliance on calibrated conventional radiocarbon ages (CRAs) and the wide range of methods that are currently used in age-depth modelling. The conversion of CRAs to calendar timescales can have variable results because of updates to the atmospheric radiocarbon calibration curve (Reimer et al., 2013). Furthermore, the calibration curves that

are used to convert conventional CRAs to calendar-equivalent time scales have variable precision through time (Reimer et al., 2013). Both factors result in changes to the slope and variability of the calibration curve, so multiple statistically significant calendar age possibilities exist for CRAs on organic material. As such, the probability distribution ranges for calibrated CRAs (and therefore their temporal uncertainty) can be large (>100 calendar years) for some time periods in the last 2000 years (Anchukaitis and Tierney, 2012).

15         The recalibration of CRAs with the most up to date calibration curve can assist in improving the representation of the uncertainty for radiocarbon dates (Blaauw, 2010). A further source of age model uncertainty comes from the interpolation or modelling of ages between dated layers, an uncertainty that decreases proportionally to the number of age constraints. Traditional age modelling approaches, including linear regression and polynomial equations, treat age uncertainty within dated horizons and interpolated ages between chronological anchors differently. Age model uncertainty

can be minimised by overlapping age probability distribution functions through a sedimentary sequence, but this is not always possible due to the cost of radiocarbon dating and availability of suitable material. Current age modelling techniques are capable of quantifying interpolation uncertainty when deriving age-depth relationships using Bayesian methods (Blaauw and Christen, 2011, Bronk Ramsay, 2009). Therefore, an added impetus of this study is to recalibrate the chronologies for the Aus2k records using Bayesian approaches to improve uncertainty estimates of future climate reconstructions.

**2.4 Age model recalibration**

In this study, one focus is to generate new age models for records that meet the PAGES2k selection criteria, providing consistency in the approach to age determination and uncertainty estimates. This study applies Bayesian age modelling across the Aus2k records, a decision that follows the initiative of the wider palaeoclimate community (e.g. Anchukaitis and Tierney, 2012, Goring et al., 2012, Hua et al., 2012). In this approach, probability functions at each dated horizon are

calculated, and then accumulation behaviour between these horizons is iteratively resampled to create thousands of possible time series. The probability distributions allow for the consideration of the non-singular nature of age calibrations for individual radiocarbon dates that form the anchors of a chronology (Hogg et al., 2013, Reimer et al., 2013).



One of the most commonly used software packages for constructing these models is 'BACON' (Bayesian ACcumulatiON histories) (Blaauw and Christen, 2011), which is run through the 'R' statistical package (R Development Core Team, 2013). Within the software, *a priori* information ('priors') informs the outcome of the statistical tests – namely the sampling interval, memory strength, and accumulation rate. Prior values can be set, based on existing knowledge of

particular archives and/or study sites (e.g. Hua et al., 2012, Goring et al., 2012), and varied by the program to determine the best fit (Blaauw and Christen, 2011). The sampling interval prior is an estimate of the realistic physical distance over which the accumulation rate can change. It is not possible to know for certain if there are shifts in accumulation rate within one physical sample, so changes are restricted to sampling intervals.

Memory strength refers to the extent to which the accumulation rate of one section relies on that of the previous

section (i.e. the degree of autocorrelation), or the likelihood of a rapid shift in the accumulation rate. For example, speleothems are observed to have minimal memory effect (Scholz et al., 2012); therefore, the memory prior should be lowered from the default (Hua et al., 2012). For the Aus2k datasets, accumulation rate has been estimated from the published age-depth model. Even if the published accumulation rate was inaccurate, the program considers how well the priors agree with the accumulation rate suggested by the data. This also allows for non-linear accumulation rates (Goring et al., 2012),

which is an important consideration in the Australasian context. For example, a rapid increase in sedimentation rate is known to correspond to European settlement in Australia (Gell et al., 2009) and changes in land cultivation practices in the Malay Archipelago (Gharibreza et al., 2013, Rodysill et al., 2012).

Anomalous dates are identified within BACON using a student's t-test to objectively exclude outliers from age model development (Blaauw and Christen, 2011). However, in this study, decisions of the original authors were upheld in

the recalibration of age models, including the exclusion of radiocarbon dates due to inversion or contamination. In addition, radiocarbon reservoir effects were applied where possible, in line with those used in the published studies. Overall, the approach used in this study maintains the specialist knowledge of the original authors, while updating records using a common statistical method for the development of age models and the calculation of uncertainty estimates.

### 3 Results

### 3.1 General attributes of Australasian palaeoclimate data

The complete compilation of Australasian records contains 661 records (Supplementary Table 1), shown in Fig. 1. The areas of high data availability occur around the coastal population centres of Australia, with density decreasing with increasing distance from cities. The Malay Archipelago has moderate, yet spatially even, coverage. New Zealand has data available across the country, with a majority of records located within the northern and eastern North Island and the western South

Island districts (Lorrey et al., 2008, Lorrey et al., 2010). Karst terrain extends in small patches along the length of both main islands in New Zealand. This has resulted in individual and composite speleothem records that have been used to infer past regional climate conditions (Lorrey et al., 2007). The most common source of terrestrial palaeoclimate information for both



New Zealand and Australia is pollen, followed by biotic microfossils. Geochemical analysis of foraminifera is the most common proxy in marine cores (Supplementary Table 1). There are a number of study sites where multiple proxies have been analysed, particularly pollen and charcoal in terrestrial cores, and $\delta^{18}O$ and Mg/Ca of foraminifer in marine cores (Supplementary Table 1).

5        As seen in Fig. 2, the length of records varies dramatically. Some lake sediment records focus on environmental change over decades (e.g. Tibby et al., 2010), while some marine cores extend into the Pleistocene (e.g. Van der Kaars and De Deckker, 2002). For many records, the youngest age is not precisely known, but the original authors assume the core top reflects collect year or 'present' (1950CE). Some records are presented as having annual or sub-annual resolution (e.g. Martin et al., 2014); however, many of these records are not constrained by annually resolved markers. These types of
10 records are regarded as non-annually resolved records in this study.

### 3.2 Results of PAGES Aus2k screening

661 Australasian sedimentary records spanning the Common Era were systematically reviewed for their suitability for reconstructing regional climate dynamics over the last 2000 years. When the PAGES selection criteria outlined in section 2.1 were applied to all Australasian records, a subset of 22 records met the stringent quality requirements (Table 1).

15        Within the screened 'high-quality' subset, the two areas with greatest geographical coverage were southeast Australia and the Indonesian Malay Archipelago (Fig. 3). This follows the pattern seen in the complete dataset (Fig. 1). Records from Indonesia are predominantly marine (n=8), but also include four terrestrial records. There are no Australian records that meet the Aus2k requirements that are located outside of the southeast region of the country; southeast Australia contains one marine core and eight terrestrial records. One of those records is physically located in New Zealand, but as a
20 record of dust accumulation, it is interpreted to be a proxy for Australian climate (Marx et al., 2009). Lacustrine microfossils are the most common terrestrial proxy in the Aus2k records, while foraminifera geochemistry is the predominant marine proxy. Average sample resolution varies; seven records have resolution less than 10 years/sample, including five lake cores, one marine core, and one speleothem. Other records resolve decadal to multi-decadal time scales.

### 3.3 Age model updates

25 The age models for all records in Table 1 were recalibrated using the techniques outlined in section 2.4. The extent of change in the minimum and maximum ages, as well as timing of anomalous events, varies within the Aus2k datasets (Table 2). Examples of recalibrated chronologies are shown in Fig. 4.

### 4 Discussion

The results presented here are the first comprehensive collection of Australasian sedimentary records that cover the last 2000
30 years. A low number of datasets are currently publically available through the data archives outlined in section 2. There is

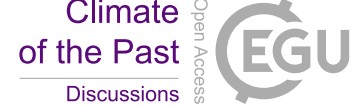

widespread spatial and temporal coverage of existing records across the geographic network (Fig. 1 and 2), with the exception of poor coverage for arid central and western Australia. Although there are proxies from the arid region that cover longer time frames over the Quaternary (Fitzsimmons et al., 2013), most available records do not have the continuity and resolution to investigate climate variability during the Common Era. The resolution of most records falls between two and

300 years (Fig. 2), which is ideal for examining variability at decadal to centennial time scales. A low number radiocarbon dates and/or low confidence in the chronologies are the most common reason for record exclusion from the high-quality dataset. Therefore, even if record length and resolution is ideal for examining climate variability over the last 2000 years, age model uncertainties means that it would be difficult to draw conclusions on the timing of events.

### 4.1 Composition of the low-resolution Aus2k dataset

The 22 records that meet the PAGES2k selection criteria provide a subset of records with robust chronologies, an identified proxy~climate variable relationship, and sufficient record length and resolution to intestigate decadal-centennial variability during the last 2000 years. As seen in Supplementary Table 1 and Fig. 2, the number of high-quality records is very low in comparison to the total number of palaeoclimate records identified and assessed. The low acceptance rate of records into the Aus2k database (3%) displays the historical focus on other aspects of Quaternary research in the Australasian region.

The geographic distribution of Australasian records displays even higher spatial biases than the complete regional database (Fig. 3). Southeast Australia has nine records, likely following the ease of access to locations proximal to major cities with research institutions, and a relative abundance of lakes and wetlands. In contrast, the availability of high quality records in Indonesia relates to global interest in the region as a 'centre of action' for numerous climate drivers including El Niño-Southern Oscillation (ENSO), the Indian Ocean Dipole (IOD), and the Australian-Indonesian Summer Monsoon
(AISM), resulting in high levels of international funding on research in this area.

New Zealand has a high number of sites that could potentially contribute more to understanding palaeoclimate of the last 2000 years; however, many extant records fail to pass the Aus2k criteria. The main reasons for New Zealand records not meeting the selection criteria are: lack of established proxy~climate variable relationship, insufficient sampling resolution, or the use of composite chronologies that require care in interpretation. Nonetheless, it is clear from the New
Zealand proxy assemblages that the presence of tephras and pollen markers can improve the chronostratigraphic constraints associated with [14]C dates as applied in many cores. Some sites also show potential to be revisited to improve age controls, sample at higher resolution, and calibrate data against instrumental records to develop a reliable climate reconstruction (Roop et al., 2015).

The subset of 22 high quality records that meet the Aus2k criteria include a range of lacustrine sediment cores,
marine cores, speleothems, and two peat cores. These archives are best suited for examining decadal to multi-decadal variability due to their accumulation rate, continuity, preservation, and dating potential. A brief description of archives and their applicability to studying climate variability during the Climate Era is provided below.





### 4.1.1 Lakes and wetlands

The majority of palaeoclimate records in Australasia are sediment cores from lake and wetland environments (Fig. 1, 3). Core are analysed for both biogenic and inorganic proxies. An important consideration of these aquatic records is site-specific factors such as sediment accumulation rate, basin morphology, and hydrological balance, which may have a strong

impact on the preservation of climate signals (Wigdahl et al., 2014). The length of lake and wetland records can vary greatly, including short, high-resolution records that focus on inter-annual changes, and long, low-resolution records that cover hundreds of thousands of years (Fig. 2).

***Ostracods***

Ostracods, small carbonate bivalve crustaceans, have been used as palaeosalinity and lake level indicators in southern

Australia for over three decades (De Deckker, 1982, Gouramanis, 2009). The regular moulting of valves throughout the organisms' lifetime means that there is a rapid response time of calcite accretion to changes in the lake environment. Originally, observed salinity tolerances of modern species were used to infer palaeosalinity (De Deckker, 1982). More recently, transfer functions have been developed to translate fossil assemblages into quantitative salinity estimates through the use of modern training sets. Training sets, expansive and detailed datasets of modern species assemblages associated

with known environmental conditions, have been developed for sites across southern Australia (De Deckker et al., 2011, Gouramanis et al., 2012, Gouramanis et al., 2010, Kemp et al., 2012). Reconstructed lake water salinity fluctuations are interpreted to reflect evaporation-precipitation (E/P) balance, assuming minimal groundwater flow into and out of the lake basin.

Geochemical analysis of ostracod valves is another proxy used to reconstruct lake chemistry and water temperature.

Monitoring of modern lacustrine systems has led to detailed understanding of controls on ostracod valve chemistry in Australian lakes. Valve Mg/Ca records both lake water Mg/Ca values and temperature (Chivas et al., 1985, Chivas et al., 1986). For much of the ostracod research in Australia, valve Sr/Ca has been interpreted as a proxy for water Sr/Ca (Chivas et al., 1985, Chivas et al., 1985) with a possible temperature influence (De Deckker et al., 1999). Recent studies argue that alkalinity is also an important influcne on valve Sr/Ca (Gouramanis and De Deckker, 2010). Stable isotope analyses are also

undertaken on the carbonate valves. Oxygen isotopes values reflect a blend of the isotopic composition and temperature of the lake water, while carbon isotopes reflect the isotopic composition of the dissolved inorganic carbon present in the lake system (see Gouramanis et al., 2010 and references therein).

Detailed ostracod-based investigations of climatic and environmental change, as well as salinity reconstructions indicative of E/P balance, are available for Blue Lake in South Australia (Gouramanis et al., 2010), Barker Swamp in

Western Australia (Gouramanis et al., 2012), Lake Keilambete and Lake Gnotuk in southern Victoria (Wilkins et al., 2013), and lakes in the northern region of Victoria (Kemp et al., 2012). Although the sample resolution at Barker Swamp and Lake Gnotuk precludes the inclusion of the sites in the Aus2k dataset, all of these sites possess a salinity reconstruction. This





diversity of proxies, as well as the high-resolution sampling of the sediment cores, has allowed for high-quality climate and environmental reconstructions based on Australian ostracod records.

### *Diatoms*

Ecological assemblages of diatoms, siliceous unicellular algae, are used to create quantitative and qualitative climatic and environmental reconstructions for lake and river systems across Australasia. Transfer functions built upon modern calibration data sets have been established for numerous aquatic variables in both estuarine (Logan and Taffs, 2013, Tibby and Taffs, 2011) and lacustrine (Gell et al., 2005) settings. Variables include conductivity/salinity (Barr, 2010, Barr et al., 2014, Gell, 1997, Saunders, 2011, Saunders et al., 2008, Saunders et al., 2007), pH (Tibby et al., 2003), and nutrients (Logan and Taffs, 2013, Tibby, 2004). Detailed studies on short time scales have examined pre-European baselines and anthropogenic impact since European settlement in the 19th century through pH (Taffs et al., 2008), salinity, and nutrient (Saunders et al., 2008) reconstructions.

Two diatom-derived conductivity reconstructions for Lake Elingamite and Lake Surprise in western Victoria are included in the Aus2k dataset (Table 1). Transfer functions for the lakes are based on 47 sites across southeast Australia, and reconstructions have been subject to multiple forms of quality control before being interpreted in a climatic framework (Barr et al., 2014). In addition, both sites have chronologies based upon radiocarbon and $^{210}$Pb dates that constrain the timing of inferred climatic shifts. Both cores have $^{210}$Pb dates from the top segment of the core to constrain the most recent rates of sediment deposition. Date density along the core, in addition to dates near the bottom of the core, leads to reduced uncertainty for age estimates for each sample (Barr et al., 2014).

Both diatom and ostracod fossils are well suited to decadal to multi-decadal reconstruction of climate over the last 2000 years because of the their short lifespan and small size. To date, most microfossil based palaeoclimate reconstructions rely upon the use of quantitative transfer functions. However, significant uncertainties have been raised with such techniques, both with biological and statistical grounds (Juggins, 2013). Nevertheless, the ecological dynamics of lakes are clearly governed by climate conditions (Battarbee, 2000), this lake microfossils maintain great potential for reconstructing past climate variability.

### *Palaeo-water isotopes*

Plant wax deuterium (δD), controlled by source water deuterium values, can be preserved in lake and marine sediment cores. Analysis of leaf n-alkanes can assist in the reconstruction of local precipitation δD values if the degree of fractionation between the leaf and water is known (Sachse et al., 2006). Two Indonesian sedimentary records within the Aus2k dataset use δD measurements as a climate proxy: one terrestrial and one marine. Konecky et al., (2013) used terrestrial plant wax compounds in lake sediments to reconstruct rainfall amount, associated with Australian-Indonesian Summer Monsoon intensity. Tierney et al, (2010) analysed lead waxes in material that had been transported offshore. The same controls on δD were assumed in both settings, with no impact by sediment transport in the marine core.

### *Lithics*



The lithic fraction of sediment cores can provide information about climate driven depositional environmental changes through time. In Australia, grain-size analyses have been used as lake-level indicators in locations where the relationship between sediment size fraction and lake levels is well understood. The most heavily studied example is Lake Keilambete in Victoria (Bowler, 1981, Bowler and Hamada, 1971, Mooney, 1997, Mooney and Dodson, 2001, Wilkins et al., 2013). The Lake Keilambete grain-size derived is very commonly used for validation and comparison of palaeocliamte records in the region. Recent high-resolution sampling of new cores, and the examination of new proxies within these cores, reinforces the interpretation of the grain size analysis (Wilkins et al., 2013). The dating density is also the highest of any of the Aus2k records, with four new accelerator mas spectrometry (AMS) radiocarbon dates and four optically stimulated luminescence (OSL) dates within the last 2000 years in the most recent chronology (Wilkins et al., 2012).

Changes in transport into a lake, reflected by accumulation rate, can be driven by climatic changes such as surface runoff or wind strength, or changed catchment conditions due to human impact (Sloss et al., 2011, Zhang et al., 2011). A rapid increase in sedimentation rates associates with European settlement is seen across Australia. Studies of wetlands and lakes across southern Australia have revealed increases from two- to eighty-fold between pre-settlement and modern day (Gell et al., 2009, Hollins et al., 2011, Sloss et al., 2011). This phenomenon is important for separating human and climatic influences on sediment cores. The lacustrine records that are available across New Zealand contain evidence for discrete sedimentation events or sedimentation variability. Signatures of acute sedimentation are interpreted as rapid in-washing of coarse loads (Lake Rotonuiaha (Wilmshurst et al., 1997); Lake Tutira (Eden and Page, 1998, Gomez et al., 2012); Round Lake (Chester and Prior, 2004)) that could be influences by climate and/or seismic variability. Some of the lithological changes observed in lake core are suggested as a response to climatically driven increases and decreases in lake level (Lake Maratoto (Green and Lowe, 1985); Lake Poukawa (McGlone, 2002a)). Other triggering factors for lake sedimentation that have been hypothesised and/or demonstrated include fluvial undercutting, anthropogenic impacts, weathering, and tectonically induced base level changes (Eden and Page, 1998).

Non-destructive analytical techniques for sediment cores are becoming more widespread in Australasian studies. These techniques, including reflectance spectroscopy (Saunders et al., 2012, Saunders et al., 2013) and X-ray florescence (Fletcher et al., 2014), can be conducted at very high (<1mm) resolution, and are easily combined with other analyses for multi-proxy studies.

Another important component of sediment deposits in Australia is the aeolian dust fraction. Dust deposition rates are a proxy for continental aridity and wind intensity, while spatial differences in deposition rates can suggest shifts in dust transportation pathways (Bowler, 1976, Hesse, 1994, Hesse and McTainsh, 2003). The method for classifying dust varies by site, and depends on the sedimentary environment. For coastal study sites, the local quartz input from sand dunes is ignored, while remaining material is assumed to be have been transported to the site via aeolian means (McGowan et al., 2008). For peat bogs, the mineral content is determined as the non-combustible fraction of dry peat mass (Marx et al., 2011, Marx et al., 2009). Mapped and modelled variations in trace element signatures and ratios of 'far-travelled' sediment can elucidate shifts in dust sources and net transport (Marx and Kamber, 2010, Marx et al., 2011, Petherick et al., 2009). Grain size and





morphology of quartz grains have also been analysed as proxies for wind speed and aeolian transport pathways (Stanley and De Deckker, 2002).

The Aus2k dataset contains two records that analyse the aeolian dust fraction in peat cores (Marx et al., 2011, Marx et al., 2009). In both records, cores were extracted from locations slightly elevated (~1m) in comparison to the surrounding

bog. This collection approach is predicted to minimise transport of local sediment and maximise atmospheric deposition. Frozen sediment cores were sampled at multi-decadal resolution. Cores from both sites were examined for trace element concentration, which allowed for reconstruction of both overall dust deposition as a proxy for continental aridity, as well as contribution of dust from specific source areas (Marx and Kamber, 2010, Marx et al., 2005). The published Snowy Mountain core had age model difficulty, with two possible age models with similar $r^2$ values (Marx et al., 2011); however, the updated

Bayesian age model clearly favoured one model over the other (Supplementary Fig. 1).

Peat bogs can be ideal sources of sediment cores because of high accumulation rates and the capture of a variety of organic and non-organic components (Barbar et al., 1994, Booth et al., 2010). Peat is a promising archive within Australasia due to the large longitudinal coverage; mires are found from tropical locations in Papua New Guinea to the sub-Antarctic islands (McGlone 2002b, McGlone et al., 2010, Whinam and Hope, 2005). Most peat-based records in Australasia contain

pollen and charcoal that may be used for palaeoclimate, palaeoecology, and paleo-fire reconstructions (Whinam and Hope 2005). Degree of humification has also been applied as a hydroclimate indicator in the late Holocene (Burrows et al., 2014, Wilmshurst et al., 2002) in cores were dating density is sufficient.

### *Pollen*

Australasia has a rich history of palynological studies, with research extending back to the 1960s (Churchill, 1960, Moar,

1967). Early research focused on reconstructing vegetation diversity at a single site, but approaches have expanded to examine broader environmental questions such as regional vegetation response to climatic shifts (Donders et al., 2007), reconstruction of a specific variable across a region (Fletcher and Thomas, 2010), recovery from episodes of disturbance such as fire (Lynch et al., 2007), and responses to human impact in the pollen catchment (Haberle et al., 2006, Horrocks et al., 2001, Leahy et al., 2005).

The investigation of the abundance and ecological assemblage of pollen spores sheds lights on palaeoecological dynamics through time, predominantly driven by changes in climate (Donders et al., 2007, Kershaw et al., 1991) and the impact by human activity (Lynch et al., 2007). There is an extensive network of well-studied sites centred on the Atherton Tablelands in northern Queensland (Haberle, 2005, Haberle et al., 2006, Kershaw, 1970, 1975, 1982, Walker, 2000), including some of the most highly cites records of Australian-Indonesian Summer Monsoon dynamics and rainforest

response to climate variability through time (Kershaw, 1994). The majority of pollen records are restricted to the peripheries of the Australian continent due to the need for water availability in the accumulation and preservation of pollen grains (Fitzsimmons et al., 2013). There have been comparisons of available pollen records for southeast Australia (D'Costa and Kershaw, 1997), and a north-south transect of high-quality pollen records along the east coast of Australia (Donders et al.,



2007). New Zealand palynology records cover the entire length of the country, and range in resolution from millennial scale to multi-decadal scale (see references in Lorrey and Bostock, 2017).

While almost all Australasian pollen reconstructions are qualitative in their approach, there have been a small number of quantitative studies. Cook and Van der Kaars (2006) comprehensively outlined early approaches (i.e. single-taxa indicators and modern analogue techniques (MAT)) and their limitations in the Australian context. The same study explored the potential of existing pollen sites to be used for construction of transfer functions, and found that regional transfer functions could be used to associate modern pollen distributions to modern hydroclimates. Herber and Harrison (2016) conducted a similar review of modern analogue techniques in Australia and suggested that, despite possible limitations in the current sampling density of the continent, MAT can be an appropriate reconstruction technique. Transfer functions have been produced for average annual temperature in Tasmania (Fletcher and Thomas, 2010) and New Zealand (Wilmshurst et al., 2007), but no quantitative reconstructions that pass the Aus2k criteria encompass the Common Era. Development of training datasets in New Zealand has historically been complicated by deforestation since 750YBP; however, a pre-deforestation dataset developed by Wilmshurst et al. (2007) mitigates this issue for future New Zealand studies.

The primary consideration in lake, wetland, or peat-derived climate archives is a clear understanding of the modern proxy~climate relationship, including the geomorphic, hydrological, geochemical, and biological response, most of which are complex and non-linear. Detailed understanding of these relationships relies on local monitoring programs or comparisons between observed proxy behaviour and instrumental records over a sufficient length of time. Development of mechanistic response models is also a priority, in order to quantify the sensitivity of certain proxies to hypothetical changes, as well as to integrate the multiple effects upon a particular palaeoclimate signal.

The most common approach in Australia is the 'calibration in space' of lake and wetland derived proxies, where lakes across an environmental gradient are sampled for the purpose of establishing modern proxy-limnology-climate relationships (Gell, 1997, Saunders, 2011). Comparisons with instrumental records form the basis of the 'calibration in time' technique, where a proxy time series is calibrated with meteorological data to produce a predictive function for quantitative reconstruction for a longer time series (Larocque-Tobler et al., 2011). Two studies within the Aus2k databset have used 20[th] century instrumental precipitation and temperature data for calibration of longer records of sediment reflection data (Saunders et al., 2013, Saunders et al., 2012). A possible limitation in the Australasian context is the assumption of a stable relationship between sediment properties and climate variables over long periods of time. Significant landscape alteration by humans has occurred during the past 2000 years, and is likely to disrupt lacustrine systems and introduce non-climatic noise into sediment cores.

### 4.1.2 Marine cores

Marine sediment cores have been collected from the entire perimeter of Australasia, some of which have accumulation rates and chronologies suited to the analysis of the last 2000 years (Fig. 1 and 3). In cores recovered in areas where sediment





incorporates terrestrial input, microfossils reflect conditions within defined horizons in the water column and organic biomarkers indicate environmental conditions in the biome of origin (Bradley, 2014). Marine sediments deposited within the last 2000 years are dated through radiocarbon methods and tephrochronology (Mohtadi et al., 2014, Oppo et al., 2009).

There is a dense concentration of marine cores in the Indonesian region, where there are large areas of shallow continental shelves (Reeves et al., 2013b). These areas remained about the lysocline through the recent past and now provide ideal sites for marine sediment core collection. Overwhelmingly the most commonly studied proxy in marine cores is the geochemistry of biogenic material. During recent millennia, oxygen isotopes of planktonic and benthic foraminifera vary as a function of both temperature and salinity (Zachos et al., 1994). For some coupled ocean-atmosphere climate modes, such as ENSO and the IOD, the coupled warm/wet and cold/dry conditions influence the $\delta^{18}O$ signature in the same direction, this

intensifying the climate signal in the oxygen isotopes (Brijker et al., 2007, Khider et al., 2011).

In places where both temperature and salinity vary on inter-annual time scales, Mg/Ca analyses allow for elucidation of the two components of the $\delta^{18}O$ signature. Mg/Ca ratios are controlled by temperature, and are independent of seawater salinity changes (Lea et al., 1999). These independent temperature determinations can then be used to remove the temperature component of $\delta^{18}O$ and isolate the salinity component. Therefore, the water temperature and water salinity can

be used independently for climate reconstructions (Elderfield and Ganssen, 2000). This has allowed the tracing of the geographic extent, dominant control, and average conditions of the Indo-Pacific Warm Pool through time (Oppo et al., 2009, Stott et al., 2004), as well as connectivity between the Pacific and Indian Oceans through the Indonesian Throughflow (Linsley et al., 2010, Newton et al., 2011). In addition, water-column dynamics have been investigated by reconstructing temperature differences between surface-dwelling and thermocline-dwelling species of foraminifera. This has led to

inferences about wind-driven mixing (Steinke et al., 2010) and properties of water masses that upwell in other regions of the globe (Khider et al., 2014).

Alternate isotopic analyses, such as the $\delta D$ interpretations outlined above, have also been applied to the marine realm (Tierney et al., 2010). In central Indonesia, one study has employed $\delta^{15}N$, controlled by basin ventilation, as a proxy for localised ENSO signatures (Langton et al., 2008). Marine proxies are well established as recorders of climate information

over long time scales. Again, sedimentation rate is the primary limitation of marine cores as recorders of Common Era climate. It is also important to separate the atmospheric and oceanographic drivers of variability in seas-surface temperatures and salinity through multi-proxy approaches.

### 4.1.3 Speleothems

Over the past half-century, speleothems have emerged as valuable sources of palaeoclimate information because of their

potential for preserving precisely dated, multi-proxy, high-resolution records of past climate change (Fairchild et al., 2006). They cover a range of temporal scales across the Holocene, from millennial (Partin et al., 2007) through to monthly to weekly resolution (Frappier et al., 2007, Treble et al., 2003, Treble et al., 2005a). Recent work indicates that Australasian speleothems offer a high-quality source of palaeoclimate information over the Common Era. Young speleothems (up to ~300



YBP) can be dated through high-precision U-Th dating methods, as long as there is sufficient uranium within the calcium carbonate matrix and the growth rate is fast enough to provide sufficient material for dating (Zhao et al., 2009). Resulting dates can have appropriate precision (+/-10-80 years) for reconstructing multi-decadal to centennial climate fluctuations (Zhao et al., 2009). Radiocarbon bomb-spike dating has also been used to date young speleothems when U-Th dating

techniques are not appropriate (Hua et al., 2012).

Speleothems fill an important geographic niche in the Australasian region, particularly for Australia, as they can be found in karst regions where a lack of standing surface water precludes the development of lacustrine records. Similar to lake cores, geochemical interpretations are site-specific, with depositional controls varying with bedrock characteristics and local climate. Moreover, the application of multiple proxies contained within speleothems can be used to narrow the range of

possible palaeoclimatic and palaeoenvironmental interpretations. The site-specific nature of many reconstructions is reliant on site monitoring, hydrologic modelling, and karst theory (Baker et al., 2014, Fischer and Treble, 2008).

The calcite mineralogy of speleothems means that they are ideal for oxygen and carbon isotope analysis and trace element concentration determinations. The complexity of site-specific controls on oxygen and carbon isotopes is recognised within speleothem studies (Lachniet, 2009, Treble et al., 2005b); as such, it is necessary to investigate the impact of

precipitation geochemistry (Fischer and Treble, 2008), processes in the aerated (vadose) zone above the cave (Dreybrodt and Scholz, 2011), and in-cave conditions (Baldini et al., 2006) on the $\delta^{18}O$ signal preserved in each speleothem. Oxygen isotopes in mid-latitude Australian speleothems primarily record rainfall amount through precipitation isotopic composition (Treble et al., 2005a) or record karst aquifer recharge frequency (Markowska et al., 2016), whereas tropical samples from Indonesia and the monsoonal region of northern Australia are mainly precipitation intensity measures (Denniston et al.,

2013, Griffiths et al., 2013). In young speleothem records, $\delta^{18}O$ values can be compared to instrumental climate data to identify correlations with temperature and rainfall (Treble et al., 2005a). Where speleothems can be sub-annually resolved, or where a significant seasonal bias can be inferred, excursions in speleothem $\delta^{18}O$ in tropical regions can be related to past cyclone events; this signal has been used to calculate cyclone occurrence over the past millennium (Haig et al., 2014). Early- to mid-Holocene $\delta^{13}C$ in a Tasmanian speleothem records was inferred to reflect bio-productivity and the resulting

fractionation in the soil (Xia et al., 2001). This is based on the extent to which atmospheric $CO_2$ is released into the soil through vegetation breakdown, and is controlled by moisture levels in the soil (Goede, 1994). The comparison of $\delta^{13}C$ analyses with trace element analyses may provide clarification of dominant controls on carbon isotopic fractionate at a given study site (Treble et al., 2005a). Treble et al. (2005a) suggest that their 20th century $\delta^{13}C$ record could partially reflect water stress on leaf stomata (Farquhar et al., 1988), based on correlation with Mg concentrations. In Indonesia speleothem records,

$\delta^{13}C$ has been found to often co-vary with Mg/Ca and Sr/Ca ratios, suggesting that these proxies respond to prior calcite precipitation (Griffiths et al., 2010, Partin et al., 2013). Prior calcite precipitation (PCP) occurs in dry conditions, when less water is transported or stored in the vadose zone, which leads to degassing of $CO_2$ into fractures in the bedrock (Fairchild et al., 2000).





In the New Zealand setting, many different factors are demonstrated to influence stable isotope signals in the speleothems (Williams et al., 2010). Previous studies, which are largely theoretical, have indicated that $\delta^{13}$C variations can be used to interpret climate changes in the form of local water balance (Lorrey et al., 2008). Including the aforementioned idiosyncrasies for cave environments and processes, local water balance has been largely assumed to represent effective

precipitation in New Zealand. Therefore, past research has exploited $\delta^{13}$C as a proxy for hydroclimatic variability related to precipitation amount. That reasoning has been employed to make inferences from speleothems about regional atmospheric circulation, which controls climate regimes and regional-scale precipitation (Lorrey et al., 2007). This synoptic approach allows the integration of co-varying, spatially heterogeneous responses of several speleothem environments to surface cliate, which is forced by orographic circulation and advection related to base climate state shifts (Lorrey et al., 2012a, Lorrey et

al., 2014, Lorrey et al., 2008).

In the Australian setting, Mg/Ca in speleothems has been shows to be a reliable recorder of effective rainfall (Fairchild and Treble, 2009, Treble et al., 2003, McDonald et al., 2004) because longer water residence times increase the Mg/Ca in speleothem drip water (Fairchild et al., 2000, Fairchild and McMillan, 2007). This relationship has been supported by comparison of recent speleothem records to instrumental datasets (Treble et al., 2003). In addition, Australian studies of

Sr/Ca has found the ratios to represent temperature-regulated fluctuations in bioproductivity above the cave (Desmarchelier et al., 2006) or a blended signal of PCP and/or growth rate, depending on the time scale examined (McDonald et al., 2004, Treble et al., 2003). Indonesian studies suggest that drip rate and residence time influence Mg/Ca and Sr/Ca ratios in locations where PCP is not a constant driver (Partin et al., 2013). P/Ca has been suggested to be a palaeorainfall proxy (Fairchild et al., 2001, Treble et al., 2003) because of the release of phosphorous from soils and subsequent transport and

incorporation associated with heavy rainfall events.

The only speleothem record in the Aus2k dataset is from Liang Luar Cave, a tropical $\delta^{18}$O record that expresses precipitation intensity above the cave (Griffiths et al., 2009). The very fine sampling interval means that the average sample resolution is high (8 years/sample); however, the average uncertainty in the recalculated age model (95% Confidence Interval = 130 years/sample) is similar to other archives.

**4.2 Discussion of age modelling approaches**

The recalibrated age-depth models developed in this study have variable agreement with the originally published models. Most studies previously published in the Australasian region have constructed chronologies through linear interpolation across median calibrated ages. Blaauw (2010) stresses the hazard of ignoring the non-singular age distribution of calibrated radiocarbon dates because of obscured error in the age depth models when date distributions are not properly acknowledged.

The Aus2k records use variable construction methods for the published chronologies and variable methods of estimating and acknowledging errors. This is the primary reason for recalibrating existing Australasian age models and comparing the outcome of updated approaches to the published models.





Chronologies for most marine sediment cores are based on linear interpolation across the set of dates. The results of BACON-derived age models moderately support the application of linear accumulation for this archive, with the ages of young/shallow samples more likely to match between the two approaches in comparison to samples from deeper in the core. The average difference between published and recalibrated ages of the youngest samples in the Aus2k marine records (n=9) is 24 years, while the average difference between oldest/closest to 1 CE is 582 years (Table 2). This is most likely related to date density through individual cores. Indonesian marine cores use the 1815 Tambora tephra as a chronological anchor, which decreases age uncertainty near the present.

Advances have been made in publically accessible statistical age model software over the past decade. The 'CLAM' (CLassical Age Modeling) software package facilitates age modelling through combining classical statistics with calibrated radiocarbon dates within a Monte-Carlo framework (Blaauw, 2010). This approach incorporates realistic probability distributions for calibrated radiocarbon dates, and can calculate age uncertainty through thousands of Monte Carlo iterations. However, the underlying statistical methods within and between classical and Bayesian approaches vary greatly (Blaauw, 2010, Blaauw and Christen, 2011, Bronk Ramsey, 2009). In the 'CLAM' software, the creator choses the type of relationship used to predict ages between chronological anchors (e.g. linear interpolation, polynomial regression, or smoothing splines), and then the program calculates likely age distributions based upon the chosen type of relationship. Blaauw (2010) suggests that Bayesian approaches construct more realistic age-depth models due to their incorporation of observed or predicted sediment behaviour; however, the simplicity of the classical approach means that it is arguably easier to construct age models, and the method remains transparent.

Two study sites in southern Australia (Barr et al., 2014) were originally published with chronologies constructed using CLAM (Blaauw, 2010). The Bayesian age models, based on the same radiocarbon and 210Pb dates, produced similar maximum and minimum ages and timing of events within the record, however notable differences are also visible, particularly for the Lake Elingamite record between 100-50YBP and between 1300-1200YBP (Supplementary Fig. 2). Differing age-depth model behaviour within these periods generally reflects differences in the degree of rigidity in the modelling approaches. That is, BACON age models appear more resistant to shifts in accumulation rate, even when low memory strength is implied within the prior settings. By contrast, CLAM allows the user to fit age models that respond more readily to changes in sediment accumulation rate. Rigidity is not a benefit or weakness of any given approach, but is a characteristic to be considered during age model construction. The literature suggests that Bayesian approaches still incorporate more site-specific information than classical methods; however, consistency in determining age uncertainties and objectivity in age modelling is arguably more important for regional comparisons (Blaauw, 2010).

The temporal coverage of the Aus2k dataset after BACON recalibration is shown in Table 2. The strongest predictors for the error envelope at the old/1CE sample are: (i) the distance between the oldest/1CE sample depth and the nearest age anchor, and (ii) the 95% confidence interval at the nearest dated horizon. The records that have the greatest departure from the published age models are those that have the smallest number chronological anchors. The international PAGES working group established a criterion of two dates for records <1000 years and three dates for records between





1000-2000 years in length. However, the outcomes of this study demonstrate that that criterion is still too relaxed for robust investigation of decadal to multi-decadal climate variability in the Common Era. For example, the age model for the one Makassar Strait record (Tierney et al., 2010) is based on six dates and one tephra layer across two cores covering the past 2000 years. The number of dates present is much higher than that suggested by PAGES2k, but dating density in this record

still results in an average 95% confidence envelope for sample ages that is greater than 100 years. Therefore, the outcome of age model recalibration in this study suggests that the criterion of two or three dates within 2000 years is not nearly stringent enough to provide robust chronologies for examining climate dynamics over sub-centennial time scales. Based on the outcomes presented here, the authors propose at least one date every 200 years, with at least one date near the top of the core, and another near the oldest sample/1CE.

It can be argued that the Bayesian methods applied in this study represent the current state of the art with regards age-depth modelling of Quaternary sediment sequences. Nevertheless, due to the ongoing development of radiocarbon calibration and modelling techniques, the age models generated here should not be viewed as permanent optimized chronologies. Radiocarbon calibration and chronology construction methods are likely to improve through time. Therefore, continuous updates of age modelling approaches could be assisted via provision of raw dates and complete chronological

metadata in publications and public data repositories.

### 4.3 Aus2k dataset limitations, potential, and recommendations

A total of 661 low-resolution palaeoclimate records from Australasia were assessed against a set of selection criteria, and a subset of 22 high-quality records were identified for future use in examining Common Era climate variability and change. Some limitations have emerged in the process of assessing the palaeoclimate dataset. These are outlined and discussed in the

following section. This review also highlights additional action items that may be undertaken to improve the data network in the Australasian region. Expansion of the Aus2k palaeoclimate data network in a geographically large, but institutionally limited region is a considerable undertaking. Therefore, a concerted community effort is required to coordinate the collection of new palaeoclimate data.

### 4.3.1 Climate variables and relationships

One reason for identifying and assessing low-resolution records in Australasia is to determine what climate variables are represented in the high-quality dataset. A distinct skew is evident for the types of climate variables currently available. New Zealand is biased towards temperature interpretations, with many of the existing low-frequency records having been constructed through multi-decadal binning of annual data (not discussed within this study). On the other hand, climate variables currently represented in the high quality Aus2k subset are heavily skewed towards hydroclimate indicators, as

reconstructed by the diverse proxies outlined in section 4.1.

Only one low-resolution terrestrial temperature reconstruction exists in the Aus2k dataset: the Duckhole Lake records (Saunders et al., 2013). A range of proxies has been used in other regions of the world to reconstruct terrestrial



palaeotemperatures in low-resolution records (Marcott et al., 2013); however, they have not yet been applied to Australasian sites at the time scales examined here. Lake temperatures have been reconstructed through chironomid transfer functions on the east coast of Australia (Chang et al., 2015), Tasmania (Rees and Cwynar, 2010), and the South Island of New Zealand (Vandergoes et al., 2008, Woodward and Shulmeister, 2007) at time scales beyond the Common Era. Modern training sets

exist for additional locations in Tasmania (Rees et al., 2008) and an additional location on South Island, New Zealand (Deiffenbacher-Krall et al., 2007). All of these transfer functions have the potential of being applied to the Common Era, this providing quantitative reconstructions from sub-tropical and temperature climate zones of Australasia.

Another temperature proxy applied to Australasian sites is the distribution of branched glycerol dialkyl glycerol tetraethers (GDGTs) in membrane lipids in sediments (Prahl and Wakeham, 1987). This proxy has been used to construct

mean annual temperature in Lake Pupuke in New Zealand (Heyng et al., 2015) and Lake Mackenzie in Australia (Woltering et al., 2014). Both temperature and aridity have been reconstructed for Onepoto Maar by combining analysis of fatty acid $\delta^{13}C$, biomass-burning biomarkers, and pollen abundances calibrated to mean annual temperatures (Sikes et al., 2013). The use of GDGTs and other lipid biomarkers has great potential for reconstructing Australasian temperatures during the Common Era, however it faces significant challenges too, including non-systematic uncertainties of ~1ºC which may

obscure the magnitude of temperature change during the last 2000 years, in addition to much greater systematic uncertainties related to the origin of the biomarkers and the type of calibration used (Woltering et al. 2014).

Borehole-derived ground surface temperature reconstructions from across the Australian continent show strong agreement with high-resolution palaeoclimate datasets (Appleyard, 2005, Cull, 1982, Huang et al., 2000, Pollack et al., 2006, Suman et al., 2016). However, the temperature estimates derived from boreholes reflect long-term, low-frequency trends that

fall outside the desired sample resolution of this study (Pollack et al., 2006).

A requirement of all palaeoclimate research is a thorough understanding of the relationship between climate variability and the geochemical, sedimentological, or biological composition of any record. To address these questions, researchers are employing long-term monitoring strategies to develop a detailed understanding of local conditions required for calibrating palaeoclimate archives (Markowska et al., 2016, Treble et al., 2003). In some cases, monitoring outcomes can

lead to some palaeoclimate records being excluded on the basis of hitherto unknown complications; Conversely, modelling the response of palaeoclimate tracers to climate change can lead to more rigorous quantitative constraints on past climates (e.g. Jones et al. 1998; 2001). Developing modern monitoring strategies should sit prominently in the collective community goal of generating more reliable regional palaeoclimate records. Site-specific monitoring projects are particularly important in the Australian region, where shortcomings in the gridded instrumental meteorological dataset are observed in rural areas

and areas of steep climatic gradients (Jones et al., 2009, Tait et al., 2006).

### 4.3.2 Chronology

The most important characteristic that a record must possess for inclusion in the 'high quality' dataset is a robust chronology, accompanied by acknowledgement and estimation of possible sources of uncertainty. Many older publications have a small





number of dates due to the high cost of radiocarbon analysis in past decades. However, there are now commercial services available low cost dating, and institutional dating operations have endeavoured to match the costs of the competitive commercial market. These developments have meant new, more cost-effective dating strategies can be employed along with renewed efforts to renewed efforts to revisit former sites to improve the original chronologies.

There are also important caveats with composite records that cover the last 2000 years (see speleothem data presented in Lorrey et al., 2008) that indicate that caution is required for accepting a record solely on the PAGES2k selection criteria. For example, previously published data may be subject to distortion because of inconsistencies in the methods used to compile overlapping series into a longer, continuous record. From this perspective, potential issues could be inherited from changes in sampling interval, as well as resultant mean and variance shifts across transitions where individual series are

spliced into longer records (Lorrey et al., 2010). A similar perspective can also be applied to irregular event-based records, such as alluvial sediments (Grant, 1985) that are bracketed by tephras and/or radiocarbon ages, or compilations of geochronology dates on moraine emplacement. In these types of cases, further work could help to evaluate the veracity of composite low-resolution records, and also determine protocols on how the development and interpretation of "master records" (e.g. isotope chronologies from several speleothems in one cave) could be improved. Inevitably, a multi-proxy

approach for corroborating interpretations is warranted.

It is possible for local factors to limit optimal dating materials or complicate the interpretation of dates (e.g. Rodysill et al., 2013). However, use of tephrostratigraphy as chronological has commonly been applied in New Zealand records (Lowe et al., 2013), has been employed in initial syntheses (Lorrey et al., 2008, Lorrey et al., 2010), and has equivalent potential in most Malay Archipelago records. The only eruption used an anchor in the Aus2k dataset is Tambora

(1815CE) in the Malay Archipelago records. Additional New Zealand tephras have been identified and characterised within the Common Era (Lowe et al., 2013), and a previously unutilised tephra in the Malay Archipelago has been characterised (Alloway et al., 2017). Identification of additional eruptions, even in the form of cryptotephra (non visible tephra layers), is a potential source of chronological tie points where datable material for radiocarbon analyses can be scarce (Gehrels et al., 2006, Gehrels et al., 2008). For example, the discovery of both basaltic and silicic glass shards in the Holocene sediments of

Lake Keilambete, Victoria, highlights the potential of using cryptotephra to constrain chronologies and correlate records across southeastern Australia and potentially between Australia and New Zealand (Smith et al., 2016).

When possible, conducting core-top radionuclide analyses, such as $^{210}$Pb and $^{137}$Cs (Appleby and Oldfield, 1978), can offer greater confidence in the age at the top of the core, as well as any significant impacts on the site by the arrival of Europeans. A robust 'young' chronology derived from the most recent section of cores may also allow comparison of lower

resolution records to instrumental observations for quantitative calibration. Future work on geochronology best practice for the Aus2k region will help to define relevant protocols and establish potential chronologic tie points.



### 4.3.3 Climate teleconnections

Previously, annually-resolved palaeoclimate records from one location within the Australasian region have demonstrated statistically significantly relationships with climatic conditions at distant locations within the same geographic area (Gallant and Gergis, 2011, Gergis et al., 2012, Ho et al., 2013). These climate teleconnections have been utilised to expand spatial

coverage of palaeoclimate reconstructions within Australasia (Ho et al., 2013). However, recent modelling and examination of documentary evidence have indicated that the climate teleconnection between Australian climate and remote drivers, such as temperatures in the western Pacific Ocean, have varied on multi-decadal to centennial time scales (Ashcroft et al., 2016, Brown et al., 2016, Hope et al., 2016, Lewis and LeGrande, 2015).

Multiple approaches for integrating Australasian palaeoclimate data have recognised the importance of

teleconnection stability and how varying teleconnection strength through time may impact on reconstruction interpretations (Gergis et al., 2012, Gallant et al., 2013, Goodwin et al., 2013, Lorrey et al., 2014, Browning and Goodwin, 2014, Gergis et al., 2016). All of the approaches are potentially limited by short length and quality of calibration time scales, uncertainties in proxy archive dating, regional biases from uneven spatial coverage, seasonal sensitivity, and in some cases multiple influences on proxy archive interpretation (i.e. potential distortion effects from other environmental processes). One

Principal Component Regression (PCR) ensemble method uses Monte Carlo simulations of analytical parameters such as principal component truncation, proxy selection, and variations in the length of calibration/verification interval to estimate reconstruction uncertainty. The ensemble spread of possible outcomes is then used as a quantitative estimate of uncertainty, which may contain a component of variability related to teleconnection instability (Gergis et al., 2012).

Another approach has applied atmospheric regimes to avoid climate proxy teleconnection dependencies. Emphasis

is placed on maximising agreement between locally derived palaeoclimate signals that are influenced by physical factors such as orography, advection, and wind stress. In that situation, local climates are assumed to respond consistently through time to synoptic-scale atmospheric circulation. In the absence of other main forcing mechanisms like volcanism, solar variability, greenhouse gases, and insolation changes, atmospheric regime frequency shifts are implicated for causing local anomalies (Goodwin et al., 2013, Jiang et al., 2013, Lorrey et al., 2014, Lorrey et al., 2007, Lorrey et al., 2008, Lorrey et al.,

2010). Past climate interpretations using this approach remain heavily reliant on modern observations, primary palaeoclimate reconstructions were proxies have been first calibrated to local climate data, palaeodata network density, and understanding how other forcing mechanisms operated and impacted local climates in the past.

One high quality record located in the South Island of New Zealand indicates dust accumulation in a peat bog may be used as a proxy for aridity on the Australian continent (Marx et al., 2009). This interpretation has been supported by

modern (1989-2001) dust provenance identification through trace element signatures (Gingele et al., 2007, Marx et al., 2005, McGowan et al., 2005). In that case, a physical process (represented by a unique dust signature) supports a plausible link to a direct, long-distance, synoptically driven transport process and, therefore, a dynamical association with remote climate forcing.




The use of 'upstream' sites could assist in improving the skill of reconstructions for a particular proxy at locations of interest. For example, in Australia, sites along the southern coast of South Australia and Victoria are impacted by a similar range of atmospheric circulation features and remote climate drivers as major cities and agricultural centres of southern Australia (Murphy and Timbal, 2008). The utilisation of high-quality sites near to, but outside, the location of interest may

lead to regional reconstructions with higher statistical skill, as they may preserve signals of large-scale circulation patterns rather than local climate features (Gallant and Gergis, 2011, Gergis et al., 2012, Gergis et al., 2016, Ho et al., 2013).

Overall, the diversity of reconstruction methods applied to the Australian region, which either do or do not estimate uncertainty associated with teleconnection instability, provides opportunities to test and compare methods, examine reconstruction assumptions, and to reconstruct climate for areas where there may be limited opportunity for data collection.

The fact that many of these approaches result in the generation of spatial fields means that there is potential to utilise their collective outputs as an ensemble, forward model, or as prospecting guidance to target and collect new records in Australasia. In particular, the resulting spatial fields and climate metrics (indices, archetypal patterns) that are able to be generated from the current range of approaches used by the Aus2k group have offered opportunities to explain how local signals in regional palaeoclimate network simultaneously arise in a dynamical context. In many cases, the different

approaches have been used as tools to explore hypotheses and reconcile apparently conflicting climate signals (spatial heterogeneity) in the Southern Hemisphere data network (Goodwin et al., 2013, Goodwin et al., 2014, Lorrey et al., 2013, Lorrey et al., 2007, Lorrey et al., 2008).

Finally, the approaches employed thus far demonstrate potential for upscaling local palaeoclimate data assemblages to be more compatible with comparisons to global climate model simulations (Ackerley et al., 2011, Lorrey et al., 2012b). A

recent reconstruction of Australian temperature over the last millennium from multiple reconstruction techniques (Gergis et al., 2016) displays a general agreement with global temperature trends (PAGES2k Consortium, 2013) in addition to region-specific timing and magnitude of temperature fluctuations identified using other approaches (see studies listed in section 4a of Gergis et al., 2016). Overall, understanding of regional idiosyncrasies is highly relevant for applying palaeoclimate data to contextualise regional responses to possible future hemispheric and global changes.

**4.3.4 Recommendations**

The following recommendations for future low-resolution palaeoclimate research of the Common Era are provided to help improve the coverage and quality of the Australasian data network:

1. The primary difficulty in establishing a basis for data comparison is a lack of publically available data. Although there are multiple public data archives available (e.g. NOAA National Centers for Environmental Information, Pangaea, Neotoma), few low-resolution records from Australasia are formally archived. It is of vital important for

continuation of data comparison in climate research that those creating records archive their existing and future data with at least one of those repositories. Future application and comparison of published data would benefit from additional metadata included in publications. Useful metadata fields include: raw geochronological data, archive





collection dates, sampling interval, temporal resolution, and the method (and data) by which the proxy~climate relationship has been established.

2. Site monitoring and climate sensitivity studies are two approaches that increase confidence in the type and strength of climate signal expressed by a proxy. It would also help improve the interpretation of previously published work. Site monitoring and the development of transfer functions to help develop frameworks for dynamical interpretation would also improve the quality of mechanistic models and quantitative climate reconstructions. Support for augmentation of global reanalysis datasets (i.e. 20CR) (Compo et al., 2011) via data rescue activities (e.g. Atmospheric Circulation Reconstructions across Earth; OldWeather, etc) (Allan et al., 2011, Allan et al., 2016, Brohan et al., 2012) to extend calibration series would provide large benefits to the palaeoclimate research community.

3. Robust chronologies are a matter of utmost importance in understanding the frequency of climate fluctuations, defining the timing of events, and testing for synchronicity of events between sites. These questions cannot be confidently answered without rigorous chronological control. Reliable chronologies require multiple dates within the past 2000 years, a date near the top and bottom of the core or near 1CE, and sufficient density of dates (at least 1 date per 200 years) across the Common Era to identify possible changes in timing within the past ~2000 years. Generation of the Bayesian age models presented here suggests that classical statistics (i.e. linear interpolation or spline smoothing) might not fully capture complex depositional pattern changes in the Australasian region.

## 5 Conclusions

Within this study, 661 non-annually resolved palaeoclimate records across the Australasian region were identified and assessed. Of these, the majority are sediment cores from lakes and wetlands, with pollen ecological assemblages and invertebrate fossil geochemistry as the most common proxies. Of the large number of records identified, a subset of only 22 records met the international PAGES2k selection criteria to be classified as 'high-quality' records. This dataset contains good examples of what characteristics are necessary for the investigation of climate variability during the Common Era. Additional records of similar quality are needed to further expand the spatial coverage and diervisty of climate variables within the Aus2k records network.

For each of the 22 records that were identified in this study, new age-depth models were constructed using consistent Bayesian age modelling. Comparison between published and BACON-derived age models suggests that age modelling has a strong influence on the timing of events within and between records. Overall, three recommendations are presented to improve the quality of future low-resolution climate reconstructions and syntheses. Public availability of data and metadata will facilitate record comparison, updates, and assessment. Thorough characterisation of proxy~climate relationships could be achieved through site monitoring, climate signal characterisation through model comparison and development and evaluation of new biological/sedimentological transfer functions. Finally, chronologies must be greatly



improved for confident characterisation of Common Era climate variability. Increased numbers of dates, core-top dating, incorporation of sediment behaviour and site idiosyncrasies in age-depth model construction, and acknowledgement of age uncertainties are all necessary for Common Era research.

The relatively low number of high quality, low-resolution records in comparison with other regions of the world highlights the progress that is needed to improve reconstructions of the climate of the past 2000 years in Australasia. However, the existing high-quality records demonstrate the potential of sites within this region to provide well-dated, high-resolution records with recognized connections to climate variables. In addition, a range of reconstruction techniques applied to other regions and time scales have the potential to expand the spatial coverage and range of climate variables in Australasia.

**Data availability**

Originally published and BACON-derived recalibrated age-depth models are archived through the NOAA paleoclimate archive.

**Author contributions**

B.C. Dixon identified and assessed the palaeoclimate Australian and Malay Archipelago records and wrote the manuscript
with the assistance of J.J. Tyler, who conducted the initial database collation. A.M. Lorrey identified and assessed the New Zealand records. I.D. Goodwin oversaw the construction of the joint report between Macquarie University and New South Wales Department of Environment, Climate Change, and Water, which was the inspiration and a major data source for this paper. J. Gergis instigated and directed the initial stages of the Aus2k low-resolution data initiative. All coauthors contributed to discussion of the content and the writing of the manuscript.

**Competing interests**

The authors declare that they have no conflict of interest.

**Special issue statement**

This is a submission for the PAGES2k special issue.

**Acknowledgements**

This is a contribution to the Past Glocal Changes (PAGES) 2k Network through the Aus2k working group. PAGES is supported by the US and Swiss National Science Foundations. BCD was supported by an Australian Postgraduate Award



and an Australian Institute of Nuclear Science and Engineering (AINSE) postgraduate award. JJT was supported by a Collaborative Research Network (CRN) fellowship and the Adelaide University Environment Institute. AML's contribution was supported by the NIWA core-funded project "Climate Present and Past" contract CAOA1701. JG was funded by Australian Research Council Project DE130100668.

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

**Figure 1. Map of all Australasian records identified. Each marker represents one record location. Archive types are represented by colour for lake/wetland (purple), marine (green), speleothem (red), and geomorphic (blue) archive types.**

**Figure 2. Metadata for all Australasian records. Data are presented for a.) Publication year for most recent presentation of each record; b.) length of time between oldest and youngest sample (where possible to calculate); and c.) sampling resolution (years/sample) of record (where possible to calculate for lake/wetland, speleothem, and marine records).**



**Table 1.** Australasian records identified as high quality, according to the PAGES2k evaluation criteria, and associated metadata. Column titles indicate: Political state where site is located (VIC=Victoria, INDO=Indonesia, SA=South Australia, NSW=New South Wales, TAS=Tasmania, NZ=New Zealand); Latitude; Longitude; Elevation (metres above sea level); Classification of archive; Source of core; Description of archive; proxies measured within core; published interpretation of climate proxy

5   (E/P=Evaporation/Precipitation balance); author of most recent publication; most recent publication year; length of record; year of oldest sample (including uncertainty); year of youngest sample (including uncertainty); average sampling resolution according to published chronology (years/sample).

**Figure 3.** Locations of high quality Aus2k records. Each marker represents one record location. Archive types are represented by colour for lake/wetland (purple), marine (green), speleothem (red), and geomorphic (blue) archive types.

10   **Table 2.** A comparison between oldest and youngest published ages and BACON-derived ages for the Aus2k high quality dataset. Published ages display published uncertainties (if available), and BACON ages display the minimum and maximum ages within the 95% confidence interval.

**Figure 4.** Examples of BACON outputs for recalibrated age-depth models for a.) Lake Elingamite; b.) Rebecca Lagoon; c.) Liang Luar Cave; d.) Marine core MD98-2181.




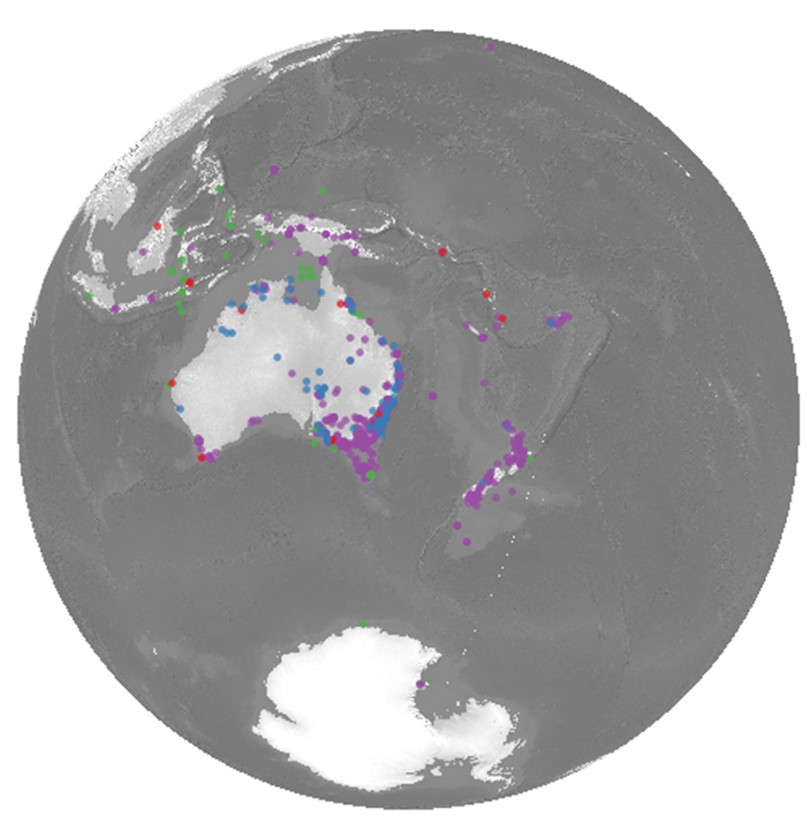



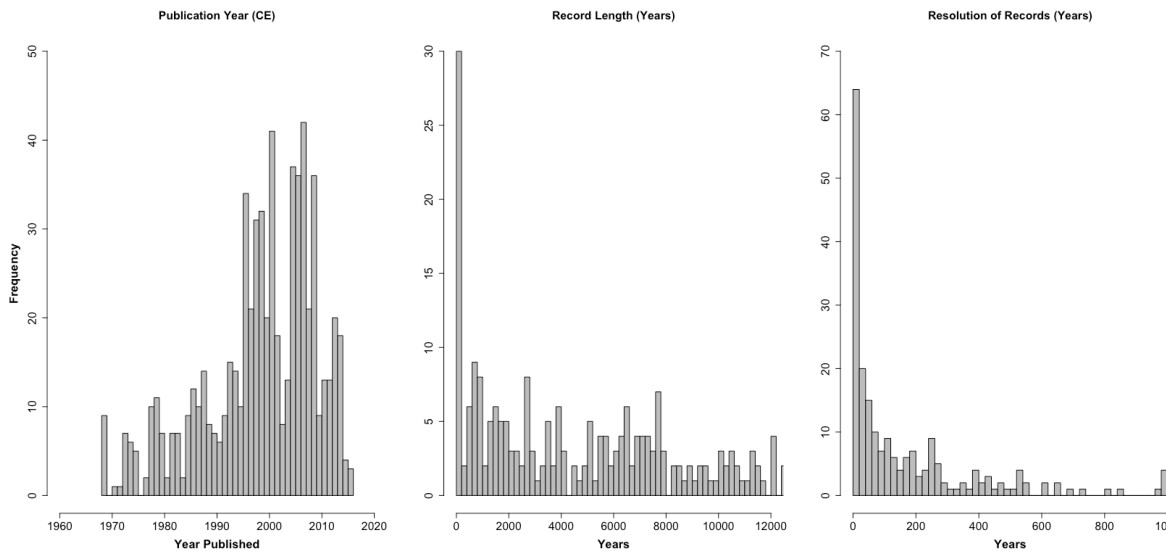



| Record Name | State | Latitude | Longitude | Elevation (m.a.s.l) | Classification | Source | Archive | Proxy | Climate Variable | Primary Author | Year Published | Number of Years in Record | Oldest Year | Youngest Year | Resolution |
|---|---|---|---|---|---|---|---|---|---|---|---|---|---|---|---|
| Lake Surprise | VIC | -38.06 | 141.92 | 93 | Lake/Wetland | Lacustrine | Diatoms | ecological assemblage | E/P | Barr | 2014 | 1440 | 1388YBP(+/-110) | -53YBP(+/-2) | 4 |
| Lake Elingamite | VIC | -38.35 | 143.00 | 121 | Lake/Wetland | Lacustrine | Diatoms | ecological assemblage | E/P | Barr | 2014 | 1552 | 1500YBP(+/-123) | -52YBP(+3) | 5 |
| Lake Lamongan | INDO | -7.98 | 113.38 | 240 | Lake/Wetland | Lacustrine | Sediments | δ18O, δ13C, Mg/Ca, laminations | E/P | Crausbay | 2006 | 778 | 731YBP | -48YBP | 1 |
| Blue Lake | SA | -37.01 | 140.01 | 24 | Lake/Wetland | Lacustrine | Diatom | ecological assemblage, Mg/Ca, Sr/Ca, Na/Ca, δ18O, δ13C | E/P, Temperature | Gouramanis | 2010 | 5690 | 5690YBP(+/-185) | 0YBP | 45 |
| Jacka Lake | VIC | -36.80 | 141.80 | 132 | Lake/Wetland | Lacustrine | Ostracods | geochemistry, ecological assemblage | E/P, Wind Strength | Kemp | 2012 | 7771 | 7721YBP(+/-1000) | -50YBP | 40 |
| Lake Lading | INDO | -8.01 | 138.31 | 324 | Lake/Wetland | Lacustrine | Plant waxes | δD | Precipitation Intensity | Konecky | 2011 | 902 | 852YBP | -50YBP | 9 |
| Upper Snowy Mountains | NSW | -36.46 | 148.30 | 1940 | Lake/Wetland | Peat | Dust | concentration, trace elements | Aridity | Marx | 2011 | 6489 | 6493YBP | 4YBP | 16 |
| Upper Ruined Hut Bog | NZ | -45.45 | 169.20 | 1420 | Lake/Wetland | Peat | Dust | concentration, trace elements | Aridity | Marx | 2009 | 7645 | 7779YBP | 134YBP | 38 |
| Lake Logung | INDO | -8.04 | 113.31 | 215 | Lake/Wetland | Lacustrine | Sediment | sedimentation rate, geochemistry | E/P | Rodysill | 2012 | 1395 | 1336YBP | -59YBP | 9 |
| Rebecca Lagoon | TAS | -41.18 | 144.68 | 8 | Lake/Wetland | Lacustrine | Sediment | reflectance | Precipitation | Saunders | 2012 | 3712 | 3654YBP | -58YBP | 12 |
| Duckhole Lake | TAS | -43.36 | 146.87 | 150 | Lake/Wetland | Lacustrine | Sediment | reflectance | Temperature | Saunders | 2013 | 908 | 850YBP | -58YBP | 2 |
| Lake Keilambete | VIC | -38.21 | 142.88 | 120 | Lake/Wetland | Lacustrine | Sediment; Ostracods | grain size; geochemistry | E/P | Wilkins | 2013 | 9596 | 9542YBP(+/-200) | -54YBP | 50 |
| Sunda Islands | INDO | -9.23 | 118.90 | -1296 | Marine | Marine | Foraminifera | δ18O, Mg/Ca | SST, Precipitation | Steinke | 2014 | 6010 | 5956YBP(+/-149) | -54YBP(+/-6) | 10 |
| MD9821-60 - Makassar Strait | INDO | -5.20 | 117.49 | -1185 | Marine | Marine | Foraminifera | δ18O, Mg/Ca | SST, Precipitation | Newton | 2006 | 836 | 946YBP | 110YBP | 10 |
| MD98-2177 - Makassar Strait | INDO | 1.40 | 119.08 | -968 | Marine | Marine | Foraminifera | δ18O, Mg/Ca | SST, Precipitation | Newton | 2011 | 1991 | 1949YBP | -42YBP | 20 |
| MD98-2176 - Western Pacific | INDO | -5.01 | 133.43 | -2382 | Marine | Marine | Foraminifera | δ18O, Mg/Ca | SST, Precipitation | Stott | 2004, 2007 | 20884 | 21010YBP | 125.7YBP | 34 |
| MD98-2181 - Morotai Basin | INDO | 6.45 | 125.83 | -2114 | Marine | Marine | Foraminifera | δ18O, Mg/Ca | SST, Precipitation | Stott | 2004, 2007 | 21866 | 21869YBP | 3YBP | 33 |
| Makassar Strait | INDO | -5.20 | 117.48 | -480 | Marine | Marine | Foraminifera | δ18O, Mg/Ca | SST, Precipitation | Oppo | 2009 | 2352 | 2316YBP | -37YBP | 10 |
| Murray Canyon | SA | -37.26 | 137.36 | -2420 | Marine | Marine | Foraminifera | δ18O, ecological assemblage | SST, ENSO activity | Moros | 2009 | 32512 | 32513YBP | 0YBP(+53) | 40 |
| Makassar Strait | INDO | -3.88 | 119.45 | -460 | Marine | Marine | Plant waxes | δD | Precipitation Intensity | Tierney | 2010 | 2329 | 2281YBP | -48YBP | 33 |
| Kau Bay | INDO | 1.00 | 127.50 | -377 | Marine | Marine | Sediments | δN | ENSO activity | Langton | 2008 | 3152 | 3332YBP | 180YBP | 50 |
| Liang Luar | INDO | -8.32 | 120.43 | 550 | Speleothem | Cave | Speleothem | δ18O, trace elements | Precipitation Intensity | Griffiths | 2009 | 12701 | 12649YBP(+/-120) | -52YBP(+/-4) | 8 |





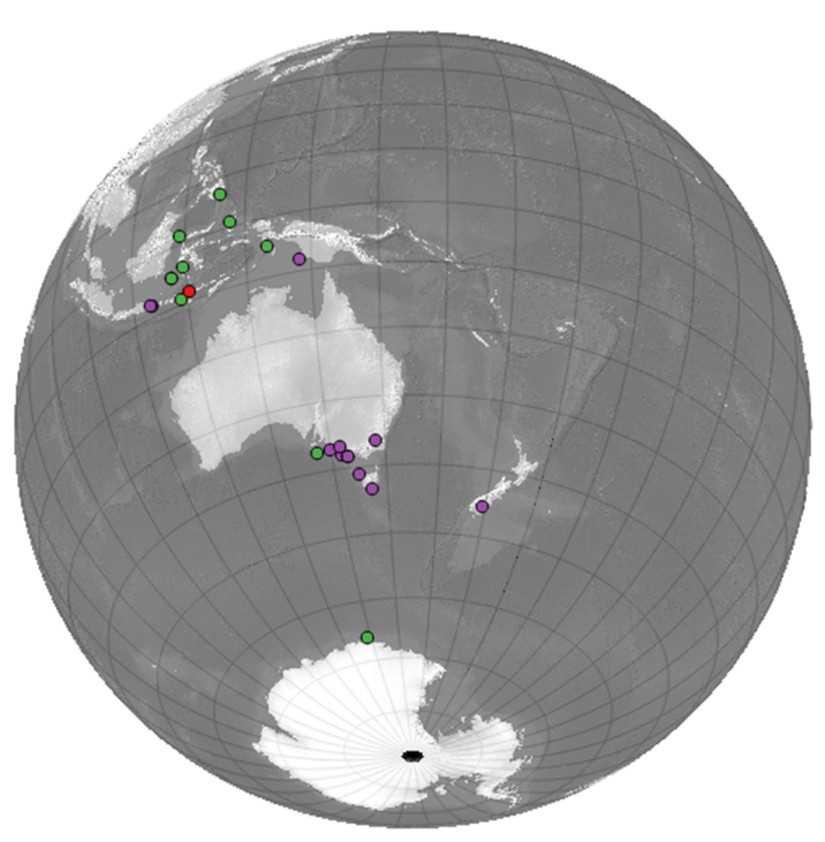





| Record Name | Oldest Published Sample Age (YBP) | Youngest Published Sample Age (YBP) | Oldest Recalibrated Sample Age (YBP) | Youngest Recalibrated Sample Age (YBP) | Difference Between Oldest Ages | Difference Between Youngest Ages |
|---|---|---|---|---|---|---|
| Lake Surprise | 1391(+/-110) | -53(+/-2) | 1396 (1259,1540) | -52 (-54,-50) | 5 | 1 |
| Lake Elingamite | 1470(+/-123) | -52(+3) | 1513 (1385,1639) | -52 (-55,-48) | 43 | 0 |
| Ranu Lamongan | 731 | -48 | 505 (480,533) | -46 (-48,-44) | 226 | 2 |
| Blue Lake | 5690(+/-185) | 0 | 5880 (5654,6134) | -47 (-52,-33) | 190 | 47 |
| Jacka Lake | 7721(+/-1000) | -50 | 6146 (5299,6945) | -23 (-44,35) | 1575 | 27 |
| Lake Lading | 1098 | -50 | 1062 (979,1148) | -49 (-52,-46) | 36 | 1 |
| Upper Snowy Mountains | 6493 | 4 | 6533 (6231,6873) | 7 (2,11) | 40 | 3 |
| Upper Ruined Hut Bog | 7779 | 134 | 6708 (6078,7420) | -8 (-46,51) | 1071 | 142 |
| Lake Logung | 1336 | -59 | 1238 (1128,1324) | -57 (-58,-55) | 98 | 2 |
| Rebecca Lagoon | 3654 | -58 | 3093 (2842,3366) | -60 (-60,-59) | 561 | 2 |
| Duckhole Lake | 810 | -58 | 802 (738,859) | -55 (-58,-52) | 8 | 3 |
| Lake Keilambete | 9357(+/-200) | -54 | 9147 (8751,9541) | -33 (-52,1) | 210 | 21 |
| Sunda Islands | 5816(+/-149) | -34(+/-6) | 5875 (5681,6027) | -33 (-50,-15) | 59 | 1 |
| MD98-2160 - Makassar Strait | NA | 110 | 1511 (1365,1656) | 21 (-25,61) | NA | 89 |
| MD98-2177 - Makassar Strait | 1949 | -42 | 2117 (1945,2318) | -25 (-48,38) | 168 | 17 |
| MD98-2176 - Western Pacific | 21010 | 126 | 20444 (19423,21458) | 115 (-46,406) | 566 | 11 |
| MD98-2181 - Morotai Basin | 21869 | 3 | 21177 (20574,21817) | -3 (-47,99) | 692 | 6 |
| Makassar Strait (Oppo et al.,2009) | 2316 | -37 | 2372 (2288,2458) | -27 (-35,-11) | 56 | 10 |
| Murray Canyon | 32430 | 0 (+53) | 29528(11261,69866) | -2(-53,117) | 2902 | 2 |
| Makassar Strait (Tierney et al., 2010) | 2281 | -48 | 2350 (2263,2437) | -37 (-40,-34) | 69 | 11 |
| Kau Bay | 3332 | 110 | 3190 (2864,3399) | 176 (-16,319) | 142 | 66 |
| Liang Luar | 1992(+/-120) | -52(+/-4) | 2002 (1936,2067) | -27 (-51,6) | 10 | 25 |

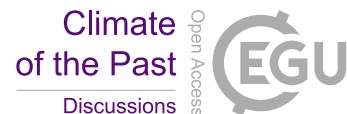



