# Peer review of "Low-resolution Australasian palaeoclimate records of the last 2000 years"

_Climate of the Past, 2017_

## Referee Comment (RC1)

**General comments**

A huge effort would have gone into compiling these records and the manuscript represents an important contribution to the PAGES2K network and the palaeoclimate community more generally.

My main comments relate to providing some additional information so readers can more easily assess old vs. new age-depth models and reproduce them, together with more of a critique about causes of differences and factors to be aware of. At present this is not possible, which detracts from the value of the paper. It would also be helpful to include direct links to each record that has been archived (see 2 below).

**Specific comments**

**Abstract**

Include briefly: where the best and worst coverage of sites is; the main reason(s) for the differences between old and new age-depth models; and summary of recommendations.

**2 Data and Methods**

As it is stated that many of the records were not publically available, were some obtained by personal communication with the original authors? If so, this should be noted in addition to NOAA and Neotoma databases. Can links to the NOAA database for each dataset in Table 1 be included? This means readers can go directly to the relevant record.

**3.3 Age model updates and 4.2 Discussion of age modelling approaches**

It would be helpful if the BACON settings for each model are provided (e.g. in supplementary material). This would ensure people can reproduce the age-depth models exactly. For example, values specified for thickness, accumulation mean and memory strength influence the output, and these are determined based on prior knowledge of the core and site as the authors highlight on page 8. If this information is not included others will not necessarily be able to reproduce the chronologies.

In Figure 4, can the original age-depth models be plotted on top of the new BACON-derived outputs (or at least side by side as done in Figure S2) so readers can see the differences between old and new? At present it is not possible to make this assessment. It would also be helpful if examples were provided of cores where new and old age-depth models are quite similar, and some examples where there are notable differences, together with a critique as to why.

While the authors state that the decisions by the original authors regarding exclusion of radiocarbon dates were upheld, was this also the case for the Lake Elingamite record (Figure S2)? The CLAM-model shows three dates (in red) that do not appear to be in the BACON model output. If these were treated as outliers by Barr et al. (2014) but still plotted, which is an option in CLAM, this should be noted in the figure caption to explain why the number of dates is different.

**4.1.1 Lakes and wetlands.**

The authors list the common factors that can have an impact on preservation of the climate signal (e.g. sediment accumulation rate, basin morphology, page 11, line 4). Human impacts, particularly since European settlement can also override potential climate signals. This can complicate the development of transfer functions/modern analogue technique models and calibration in time. This is highlighted in the 'lithics' section, but is relevant to biological proxies too.

Page 14, lines 8-10 and Supplementary Fig. 1: State which age model the BACON-derived one supports and why.

Page 15, lines 14-19: This paragraph is better suited near the start of the section before discussing the different proxies as it is general. Some references would be beneficial, in particular in relation to proxies being complex and non-linear. In addition, using a multi-proxy approach is important for being able to potentially discern climate vs. human impact vs. within lake signals.

Page 15, line 24: von Gunten et al. (2012) is an additional reference as it describes the calibration in time approach using case studies based on biogeochemistry (von Gunten L, Grosjean M, Kamenik C, Fujak M, Urrutia R (2012) Calibrating biogeochemical and physical climate proxies from non-varved lake sediments with meteorological data: methods and case studies. J Paleolimnol DOI 10.1007/s10933-012-9582-9).

Page 19, lines 4-7: the difference between the oldest/closest to 1 CE dates are also likely due to the amount of extrapolation between it and the previous 14C date, not just the density of dates through the core. In addition to having at least one date near the top of the core (line 8), ideally 210Pb and/or 137Cs would be used if the sedimentation rate is fast enough.

**Figures and Tables**

**Figures 1 and 3**

What symbol marks peat records?

**Tables**

It is not clear how studies are ordered, which makes it hard to search through them (also applies to Table S1)

**Table 2**

Just presenting the difference between top and base years in the old compared to new chronologies is useful, but does not necessarily illustrate the actual differences between the age-depth models. For example, if the lowermost 14C date is above the bottom of the core, which means the age-depth models are extrapolated, this may lead to a larger apparent difference in ages than might be the case for most of the core. To address this, figures of each site with old and new age-depth models could be included. All new age-depth model figures should already be available as part of BACON output. Doing this means the settings could be incorporated into each figure (see comment 3.3 above). Ideally the original age-depth models would be plotted on top to best illustrate differences and similarities. If this is not possible, then at least provide them side by side. This would help readers assess the differences for themselves, identify common patterns and assist their decision making when investigating these records and developing chronologies for other sites.

**Technical corrections**

There are a number of typographical errors in the text. I have listed the ones I found, but recommend the authors do a thorough check. This includes the order of references within the text, which are not always consistent (e.g. Marx et al., 2011, Marx et al., 2009, page 13, line 32).

Page 1, lines 15-16: A high quality subset of 22 records across Australasia met the criteria and  **were** endorsed for subsequent analyses

Page 2, line 30: Low-resolution sedimentary  **archives** available within Australasia include lacustrine…

Page 3, line 13: state the most recent year of publication of the records so it is clear until when the database is up to date. This is important because new records are being published (e.g. comments by Rouillard).

Page 4, line 12: a reference is needed at the end of the sentence The Australasian region includes tropical Southeast Asia because of the dynamical influences of the Indo-Pacific region on the Australasian monsoon. As two Antarctic sites appear to be included, the reason why stated.

Page 4, line 19: 'Reasonable' was defined  **by** PAGES2k as containing at least one…

Page 5, line 1: …approach for the creation of age models  presented in this study…

Page 5, line 8:  **Of** these 241 records…

Page 5, line 27: Should temperature be temperate?

Page 6, line 27: …residence time within a radiocarbon sample…

Page 7, lines 26-27: Sentence is not necessary – In this study, one focus is to generate new age models for records that meet the PAGES2k selection criteria, providing consistency in the approach to age determination and uncertainty estimates.

Page 7, lines 27-29: Sentence is not necessary as it overlaps with the end of the previous paragraph. Combine it with the previous paragraph so the references are included. This study applies Bayesian age modelling across the Aus2k records, a decision that follows the initiative of the wider palaeoclimate community (e.g. Anchukaitis and Tierney, 2012, Goring et al., 2012, Hua et al., 2012).

Page 9, lines 13-14: Sentence not necessary – 661 Australasian sedimentary records spanning the Common Era were systematically reviewed for their suitability for reconstructing regional climate dynamics over the last 2000 years.

Page 9, lines 21-22: Sentence is a repeat of the previous section – Lacustrine microfossils are the most common terrestrial proxy in the Aus2k records, while foraminifera geochemistry is the predominant marine proxy.

Page 10, line 5: Missing word: A low number **of** radiocarbon dates…

Page 10, line 11: …resolution to  **investigate** decadal…

Page 10, line 15 paragraph: This appears to be a contradiction to the start of the Discussion where the authors comment there is widespread spatial and temporal coverage of existing records across the geographic network (page 10, line 1). Please reword to clarify what is meant.

Page 10, lines 18-19: …climate drivers including **the** El Niño-Southern Oscillation (ENSO),  Indian Ocean Dipole (IOD), and  Australian-Indonesian Summer Monsoon…

Page 10, line 32:  **Common** Era

Page 11, line 2: Core**s**

Page 11, line 23: Chivas et al., 1985

Page 11, line 25: Oxygen isotope values…

Page 12, line 5: Transfer functions  **developed from** modern calibration…

Page 12, line 20: … because of  their…

Page 12, line 22: …both  **on** biological and statistical grounds…

Page 12, lines 22-23: this sentence is misleading (Nevertheless, the ecological dynamics of lakes are clearly governed by climate conditions) because the ecological responses in some lakes are influenced more by humans than climate (e.g. damming), or changes in the catchment that are not climate-driven (e.g. land clearing), or could just vary internally.

Page 12, line 23: …this lake microfossils… – should 'this' be 'thus'?

Page 12, line 31: … analysed lead waxes… – should 'lead' be 'leaf'?

Page 13, line 5: Does not make sense – The Lake Keilambete grain-size derived is very commonly used for validation and comparison of  **palaeoclimate** records in the region

Page 13, line 8: …accelerator mas**s** spectrometry…

Page 13, line 18: …could be influence**d** by climate…

Page 13, line 19: …lake core**s** are…

Page 13, line 31: …assumed to  have been…

Page 14, lines 11-17: This paragraph does not belong here as is discusses an archive, not a proxy. It would be more suitable at the start of section 4.1.1. and the title adjusted to include peat.

Page 14, line 27: …impact**s**…

Page 14, line 29: …highly cite**d** records…

Page 15, line 7: Herber**t**

Page 16, line 9: …this intensifying… – should 'this' be 'thus'?

Page 16, line 26: …in seassurface…

Page 17, line 24: …record

Page 18, line 8: cli**m**ate

Page 18, line 11: shows**n**

Page 19, line 33: …number **of** chronological…

Page 20, line 20: …demonstrate that  **this** criterion…

Page 20, line 10: …with regards **to** age-depth…

Page 20, lines 17-18: Sentence is not necessary

Page 20, line 32: record

Page 21, line 6: …this providing… – should 'this' be 'thus'?

Page 21, line 6: D**ei**ffenbacher-Krall et al., 2007 should be Di**ef**fenbacher

Page 21, line 9: …used to **re**construct…

Page 21, line 25: …unknown complications; **c**onversely, modelling…

Page 22, line 17: Sentence starting 'However' is not clear.

Page 22, line 19: …used **as** an anchor…

Page 22, lines 27-29: Provide an Australian and ideally New Zealand reference at the end of the sentence.

Page 23, line 12: …limited by **the** short length…

Page 23, line 26: …reconstructions w**h**ere proxies…

Page 24, lines 2-4: Sentence starting 'For example' is not clear.

Page 24, line 14: …signals in **a** regional palaeoclimate…

Page 24, line 30: …vital importance for…

Page 25, line 4: …strength of **a** climate…

Page 25, line 24:  **diversity**

Page 25, line 25: …Aus2k  network.

Page 25, line 31: …model comparison**,** and…

Page 26, line 3: 'for Common Era research' is not necessary

Page 26, lines 6-7: Do the authors mean 'high-resolution' or 'low resolution'?

Page 26, line 7: …recogni**s**ed…

**Supplementary material**

Figure S1: Make axes the same units and scales

Figure S2: Make axes the same units and scales

**References**

There are a number of references missing from the reference list or text. Below are the ones I found. I recommend the authors check through all text, references and supplementary material to make sure all references are included and there are no typographical errors.

**The following references are missing from the reference list:**

Browning and Goodwin 2014

De Deckker et al. 2011

Emile-Geay and Eshleman, 2013

Gingele et al., 2007

Goring et al. 2012

Gouramanis et al. 2010

Grant 1985

Jones et al. 1998, 2001

Kershaw 1982

Martin et al. 2014

R Development core team 2013

Schaefer et al., 2009

**In the reference list:**

Typo: Bowler, J. M. & Hamada, T. 1971. Late Quaternary stratigraphy and radiocarbon chronology of water level fluctuations in Lake Keilambete, Victoria. Nature, 232, 330-**&.**

**Please recheck the references for the correct format of surnames. Below are the ones I noticed.**

D'costa should be D'Costa

McTainsh (Hesse, P. P. & **Mctainsh**, G. H. 2003. Australian dust deposits: modern processes and the Quaternary record. Quaternary Science Reviews, 22, 2007-2035)

Mcdonald should be McDonald

Mcfadgen should be McFadgen

Mcglone should be McGlone

Mckay should be McKay

McMillan (Fairchild, I. J. & Mc**m**illan, E. A. 2007. Speleothems as indicators of wet and dry periods. International Journal of Speleology, 36, 69-74)

LeGrande (Lewis, S. C. & Le**g**rande, A. N. 2015. Stability of ENSO and its tropical Pacific teleconnections over the Last Millennium. Climate of the Past, 11, 1347-1360)

---

## Short Comment (SC1) · 21 Mar 2017

Dear authors,

Considering the relative paucity of records in the Western and Northern parts of Australia, I would like bring to your attention the following recently published low resolution records for these regions, with respective references, for addition to Figure 1 & Figure 2.

Black Springs LakeWetland McGowan et al., 2012 King River LakeWetland Proske et al., 2014 Fortescue Marsh LakeWetland Rouillard et al., 2016a, b

McGowan, H. et al. (2012) Evidence of ENSO mega-drought triggered collapse of prehistory Aboriginal society in northwest Australia. Geophysical Research Letters 39,

L22702, 1–5

Proske, U. et al. (2014) A Holocene record of coastal landscape dynamics in the eastern Kimberley region, Australia. Journal of Quaternary Science 29, 163–174

Rouillard, A. et al. (2016a) Evidence for extreme floods in arid subtropical northwest Australia during the Little Ice Age chronozone (CE 1400–1850). Quaternary Science Reviews 144, 107 – 122

Rouillard, A. et al. (2016b) Interpreting vegetation change in tropical arid ecosystems from sediment molecular fossils and their stable isotope compositions: A baseline study from the Pilbara region of northwest Australia. Palaeogeography, Palaeoclimatology, Palaeoecology 459, 495–507

Kind Regards,

Alexandra Rouillard, PhD alexandra.rouillard@snm.ku.dk; +45 52 73 25 50 MSCA Postdoctoral Fellow Centre for GeoGenetics Natural History Museum of Denmark University of Copenhagen Øster Voldgade 5-7, 1350 Copenhagen K, Denmark

---

## Referee Comment (RC2) · Anonymous Referee #2 · 18 May 2017

This is a very clearly written manuscript, making a convincing case for compiling multiple low-resolution archives of past environmental/climate change in the Australasian region. The paper could be useful for future palaeo-studies in the region and could inspire research teams to produce similar compilations for other regions.

Which calibration curve was used for the terrestrial sites, SHCal13? Make this clear within the methods. For the marine sites, how were marine dR values and their uncertainties estimated, e.g. using http://calib.org/marine/ ? Which data-points were used to estimate dR values for each site? Please provide this information as supplementary information or at your NOAA archive, so that others can replicate your findings.

p1 line 16, what are progressive Bayesian techniques?

p2 line 4, but one could argue that during this recent time, human impact might have

affected more of the proxy records. Could this potentially be a problem in some of your sites?

p6 line 12, don't forget to list the error associated with having non-dated levels, and thus requiring an age-model that provides realistic estimates of uncertainties (as you explain later, on p7 lines 16-24). Perhaps cite Bennett, K.D. 1994 (The Holocene 4, 337-348), Telford et al. 2004 (Quat. Sci. Rev. 23, 1-5), and Trachel & Telford 2016 (The Holocene doi:10.1177/0959683616675939).

Perhaps cite Flantua et al 2016 (Climate of the Past 12, 387-414) for another recent compilation of regional chronologies.

p8, line 18, Bacon does not exclude outliers but deals with them through using student-t distributions for all dates as default (not student-t tests) - these distributions look much like normal distributions but have wider tails. As a result even dates that seem outlying to our eyes (i.e. lying far away from the model and neighbouring dates) will often still fit the age-model (probability distribution >0 at the age-model at the depth of said date).

Language

p12 lines 22-24, check sentence

p13 line 12, associated & line 18, influenced

p17 line 29, Indonesian

p18 line 11, shown

p22 line 4, renewed efforts to renewed efforts to

p25 line 24, diversity

p26 line 25, Past Global Changes (not glocal)

---

## Short Comment (SC2) · 22 May 2017

The PAGES Data Stewardship Integrative Activity seeks to advance best practices for sharing data generated and assembled as part of all PAGES-related activities. As part of this activity, a team of reviewers has been constituted for the "Climate of the Past 2000 years" Special Issue. The data team is reviewing the data handling within each of the CP-Discussion papers in relation to the CP data policy and current best practices. The team has identified essential and recommended additions for each paper, with the goal of achieving a high and consistent level of data stewardship across the 2k Special Issue. We recognize that an additional effort will likely be required to meet the high level of data stewardship envisaged, and we appreciate the dedication and contribution of the authors. This includes the use of Data Citations (see example in supplement). We ask authors to respond to our comments as part of the regular

open interactive discussion. If you have any questions about PAGES Data Stewardship principles, please contact any of us directly.

Best wishes for the success of your paper,

2k Special Issue Data Review Team (Darrell Kaufman, Nerilie Abram, Belen Martrat, Raphael Neukom, Scott St. George) and ex-officio team members (Marie-France Loutre, Lucien von Gunten)

Essential additions for this paper:

(1) Expand the "Data Availability" section to include a URL to a landing page for the data compilation in this paper. The landing page should list the 22 individual datasets that were selected for this study (Table 1) plus those that were considered but not selected (Table S1).

(2) Add Data Citations/URLs (in addition to publication citations) for each of the 22 datasets selected for this compilation. For those data not already in a persistent public repository, submit essential metadata along with the proxy time series itself and add the Data Citation/URL in Table 1.

(3) Include the 14C and other age control for each of the 22 records, plus the age ensembles (a primary outcome of this study), within the publicly archived files (LiPD format is recommended).

(4) If any of the records used in this study were also used in previous PAGES 2k databases (temperature or isotopes), please include cross references to those IDs in Table 1.

Strongly recommended:

(5) Table S1 is a major resource for the paleoclimate community. Its value would be increased by: (a) Stating the criterion (or criteria) that was not met for each of the records that did not make the final cut. This will enable future users to easily cull the

records that meet other criteria (e.g., lower resolution). (b) Naming the archive and proxy type for each record (as in Table 1). (c) Replacing the "NA" with the actual data or explain the purpose/meaning of the "NA".

Please also note the supplement to this comment:
http://www.clim-past-discuss.net/cp-2017-31/cp-2017-31-SC2-supplement.pdf

---

## Author Comment (AC2) · 23 Jun 2017

Dear Dr. Kaufman and 2k Special Issue Data Review Team,

Thank you for your comments on 'Low-resolution Australasian palaeoclimate records of the last 2000 years'. The authors look forward to working to ensure correct data archiving is achieved.

Essential additions for this paper: (1) Expand the "Data Availability" section to include a URL to a landing page for the data compilation in this paper. The landing page should list the 22 individual datasets that were selected for this study (Table 1) plus those that were considered but not selected (Table S1).

Response: The following URL will become active upon publication of this manuscript:

ftp://ftp.ncdc.noaa.gov/pub/data/paleo/pages2k/dixon2017australasia/.

Age control information, the original published age-depth model, the updated age-depth model, and the previously published time series for each of the 22 'high quality' data sets will be available at this URL.

Raw datasets for each of the publications in table S1 have not been collected. If the data team would prefer, it is possible to include table S1 in spreadsheet form in the online data folder.

(2) Add Data Citations/URLs (in addition to publication citations) for each of the 22 datasets selected for this compilation. For those data not already in a persistent public repository, submit essential metadata along with the proxy time series itself and add the Data Citation/URL in Table 1.

Response: A column will be added to table 1, which will contain data URL(s) for each of the 22 PAGES Aus2k datasets. Datasets not currently available in public archives have been submitted to NOAA paleoclimate, and the URLs will be included in table 1.

(3) Include the 14C and other age control for each of the 22 records, plus the age ensembles (a primary outcome of this study), within the publicly archived files (LiPD format is recommended).

Response: Currently published age control information is available at the individual data URLs mentioned in the previous comment. The BACON age-depth models will be available at ftp://ftp.ncdc.noaa.gov/pub/data/paleo/pages2k/dixon2017australasia/ upon publication of this manuscript.

(4) If any of the records used in this study were also used in previous PAGES 2k databases (temperature or isotopes), please include cross references to those IDs in Table 1.

Response: Data IDs of records used in previous PAGES2k databases will be included in a 'Data ID' column in table 1.

Strongly recommended:

(5) Table S1 is a major resource for the paleoclimate community. Its value would be increased by: (a) Stating the criterion (or criteria) that was not met for each of the records that did not make the final cut. This will enable future users to easily cull the records that meet other criteria (e.g., lower resolution). (b) Naming the archive and proxy type for each record (as in Table 1). (c) Replacing the "NA" with the actual data or explain the purpose/meaning of the "NA".

Response: Table S1 will be updated to include the archive and proxy for each record. The primary reason for exclusion of records will be provided. 'NA' was used when it was not possible to calculate the field (i.e. resolution) and/or when it was not possible to access the dataset. This will be clarified in the table caption.

Please also note the supplement to this comment: http://www.clim-past-discuss.net/cp-2017-31/cp-2017-31-SC2-supplement.pdf

---

## Author Comment (AC3) · 23 Jun 2017

General comments A huge effort would have gone into compiling these records and the manuscript represents an important contribution to the PAGES2K network and the palaeoclimate community more generally. My main comments relate to providing some additional information so readers can more easily assess old vs. new age-depth models and reproduce them, together with more of a critique about causes of differences and factors to be aware of. At present this is not possible, which detracts from the value of the paper. It would also be helpful to include direct links to each record that has been archived (see 2 below).

Response: This identification and assessment of non-annually resolved Australasian

palaeoclimate records has been underway since 2009, and has been the focus of both post-doctorate research and a PhD project. We are pleased to present the outcomes of more than eight years of work, and hope that the scientific community can benefit. Original and updated age models and datasets will be available through NOAA World Data Center at:

ftp://ftp.ncdc.noaa.gov/pub/data/paleo/pages2k/dixon2017australasia/

This link will become active upon publication of this manuscript.

Specific comments

Abstract Include briefly: where the best and worst coverage of sites is; the main reason(s) for the differences between old and new age-depth models; and summary of recommendations.

Response: The abstract will be updated to include a brief summary of i) areas of densest and sparsest record coverage, ii) the main reason(s) for differences between initially published and updates age-depth models, and iii) recommendations for future improvement of palaeoclimate research and records in Australasia.

2 Data and Methods As it is stated that many of the records were not publically available, were some obtained by personal communication with the original authors? If so, this should be noted in addition to NOAA and Neotoma databases.

Response: Sources of pre-existing records will be updated to include 'a general inquiry to Australasian Quaternary Association members' and 'personal communication with authors'

Can links to the NOAA database for each dataset in Table 1 be included? This means readers can go directly to the relevant record.

Response: Links to the NOAA URL for originally published datasets will be included in Table 1. A link to the comparison of originally published and updated age models will

be included in the 'Data availability' section at the end of the manuscript.

3.3 Age model updates and 4.2 Discussion of age modelling approaches It would be helpful if the BACON settings for each model are provided (e.g. in supplementary material). This would ensure people can reproduce the age-depth models exactly. For example, values specified for thickness, accumulation mean and memory strength influence the output, and these are determined based on prior knowledge of the core and site as the authors highlight on page 8. If this information is not included others will not necessarily be able to reproduce the chronologies.

Response: A supplement of all code and settings used to construct updated age models will be included in the supplementary information, for the purpose of transparency and reproducibility of methods presented in this study

In Figure 4, can the original age-depth models be plotted on top of the new BACON-derived outputs (or at least side by side as done in Figure S2) so readers can see the differences between old and new? At present it is not possible to make this assessment. It would also be helpful if examples were provided of cores where new and old age-depth models are quite similar, and some examples where there are notable differences, together with a critique as to why.

Response: Figure 4 will be updated to show the four example age-depth models side-by-side with the originally published age models so that differences can be observed. Each pair of old/new age models will be provided in the supplementary information, so that readers can see examples of similar and divergent age models. Further discussion of reasons for differences in age model outcomes will be included in section 4.2

While the authors state that the decisions by the original authors regarding exclusion of radiocarbon dates were upheld, was this also the case for the Lake Elingamite record (Figure S2)? The CLAM-model shows three dates (in red) that do not appear to be in the BACON model output. If these were treated as outliers by Barr et al. (2014) but still plotted, which is an option in CLAM, this should be noted in the figure caption to

explain why the number of dates is different.

Response: The dates presented in red in Barr et al., (2014) have indeed been excluded by this study because of their previous identification as outliers. This will be clarified in the caption of figure S2.

4.1.1 Lakes and wetlands.

The authors list the common factors that can have an impact on preservation of the climate signal (e.g. sediment accumulation rate, basin morphology, page 11, line 4). Human impacts, particularly since European settlement can also override potential climate signals. This can complicate the development of transfer functions/modern analogue technique models and calibration in time. This is highlighted in the 'lithics' section, but is relevant to biological proxies too.

Response: The sections on biological proxies will be updated to appropriately communicate the impact of site-specific feature and land use change on biological proxies in Australasia. Relevant supporting literation will also be included.

Page 14, lines 8-10 and Supplementary Fig. 1: State which age model the BACON-derived one supports and why.

Response: This section will be clarified and expanded upon to indicate which published model is preferred, as indicated by the BACON-derived age model, and why BACON is a useful tool for choosing appropriate age models.

Page 15, lines 14-19: This paragraph is better suited near the start of the section before discussing the different proxies as it is general. Some references would be beneficial, in particular in relation to proxies being complex and non-linear. In addition, using a multi-proxy approach is important for being able to potentially discern climate vs. human impact vs. within lake signals.

Response: This paragraph will be shifted to the beginning of the section (page 11, line 7), and will be edited to prevent repetition with the existing introductory paragraph

(page 11, lines 2-7). Supporting literature will be added to support the complex and non-linear nature of palaeoclimate proxies in lake and wetland environments.

Page 15, line 24: von Gunten et al. (2012) is an additional reference as it describes the calibration in time approach using case studies based on biogeochemistry (von Gunten L, Grosjean M, Kamenik C, Fujak M, Urrutia R (2012) Calibrating biogeochemical and physical climate proxies from non-varved lake sediments with meteorological data: methods and case studies. J Paleolimnol DOI 10.1007/s10933-012-9582-9).

Response: Information from this reference will be incorporated into page 15, line 24, and the paper will be added to the reference list.

Page 19, lines 4-7: the difference between the oldest/closest to 1 CE dates are also likely due to the amount of extrapolation between it and the previous 14C date, not just the density of dates through the core. In addition to having at least one date near the top of the core (line 8), ideally 210Pb and/or 137Cs would be used if the sedimentation rate is fast enough.

Response: The degree of interpolation will be discussed in the possible reasons for large chronological uncertainties. References will be supplied to support this point. The authors agree that 210Pb and/or 137Cs would be ideal for constraining the top of sediment cores in Australasia, and these radionuclides have has been used in previous studies. The use of 210Pb dating for constraining upper-core ages is included in the 'chronology' section of the discussion (page 22, line 27).

Figures and Tables

Figures 1 and 3 What symbol marks peat records?

Response: Peat records are included in the lake/wetland category, which is represented by the purple markers. The figure captions will be changed to clarify the types of proxies included within each archive category.

Tables It is not clear how studies are ordered, which makes it hard to search through

them (also applies to Table S1)

Response: Records within all tables will be reordered by archive type (i.e. lake/wetland, marine, speleothem, marine), then by state/country, for the sake of easy searching by readers.

Table 2 Just presenting the difference between top and base years in the old compared to new chronologies is useful, but does not necessarily illustrate the actual differences between the age-depth models. For example, if the lowermost 14C date is above the bottom of the core, which means the age-depth models are extrapolated, this may lead to a larger apparent difference in ages than might be the case for most of the core. To address this, figures of each site with old and new age-depth models could be included. All new age-depth model figures should already be available as part of BACON output. Doing this means the settings could be incorporated into each figure (see comment 3.3 above). Ideally the original age-depth models would be plotted on top to best illustrate differences and similarities. If this is not possible, then at least provide them side-by-side. This would help readers assess the differences for themselves, identify common patterns and assist their decision making when investigating these records and developing chronologies for other sites.

Response: A comparison figure displaying overlain old and new age models for each of the 22 sites will be constructed and included in the supplementary material. Table 2 will be removed and readers will be directed to the supplementary information.

Technical corrections There are a number of typographical errors in the text. I have listed the ones I found, but recommend the authors do a thorough check. This includes the order of references within the text, which are not always consistent (e.g. Marx et al., 2011, Marx et al., 2009, page 13, line 32).

Page 1, lines 15-16: A high quality subset of 22 records across Australasia met the criteria and they are were endorsed for subsequent analyses

Page 2, line 30: Low-resolution sedimentary archived archives available within Australasia include lacustrine. . .

Page 3, line 13: state the most recent year of publication of the records so it is clear until when the database is up to date. This is important because new records are being published (e.g. comments by Rouillard).

Page 4, line 12: a reference is needed at the end of the sentence The Australasian region includes tropical Southeast Asia because of the dynamical influences of the Indo-Pacific region on the Australasian monsoon. As two Antarctic sites appear to be included, the reason why stated.

Page 4, line 19: 'Reasonable' was defined as by PAGES2k as containing at least one. . .

Page 5, line 1: . . .approach for the creation of age models in presented in this study. . .

Page 5, line 8: Or Of these 241 records. . .

Page 5, line 27: Should temperature be temperate?

Page 6, line 27: . . .residence time within a radiocarbon samples. . .

Page 7, lines 26-27: Sentence is not necessary – In this study, one focus is to generate new age models for records that meet the PAGES2k selection criteria, providing consistency in the approach to age determination and uncertainty estimates.

Page 7, lines 27-29: Sentence is not necessary as it overlaps with the end of the previous paragraph. Combine it with the previous paragraph so the references are included. This study applies Bayesian age modelling across the Aus2k records, a decision that follows the initiative of the wider palaeoclimate community (e.g. Anchukaitis and Tierney, 2012, Goring et al., 2012, Hua et al., 2012).

Page 9, lines 13-14: Sentence not necessary – 661 Australasian sedimentary records spanning the Common Era were systematically reviewed for their suitability for reconstructing regional climate dynamics over the last 2000 years.

Page 9, lines 21-22: Sentence is a repeat of the previous section – Lacustrine micro-fossils are the most common terrestrial proxy in the Aus2k records, while foraminifera geochemistry is the predominant marine proxy.

Page 10, line 5: Missing word: A low number of radiocarbon dates. . .

Page 10, line 11: . . .resolution to intestigate investigate decadal. . .

Page 10, line 15 paragraph: This appears to be a contradiction to the start of the Discussion where the authors comment there is widespread spatial and temporal coverage of existing records across the geographic network (page 10, line 1). Please reword to clarify what is meant.

Page 10, lines 18-19: . . .climate drivers including the El Niño-Southern Oscillation (ENSO), the Indian Ocean Dipole (IOD), and the Australian-Indonesian Summer Monsoon. . .

Page 10, line 32: Climate Common Era

Page 11, line 2: Cores

Page 11, line 23: Chivas et al., 1985, Chivas et al., 1985

Page 11, line 25: Oxygen isotopes values. . .

Page 12, line 5: Transfer functions built upon developed from modern calibration. . .

Page 12, line 20: . . . because of the their. . .

Page 12, line 22: . . .both with on biological and statistical grounds. . .

Page 12, lines 22-23: this sentence is misleading (Nevertheless, the ecological dynamics of lakes are clearly governed by climate conditions) because the ecological responses in some lakes are influenced more by humans than climate (e.g. damming), or changes in the catchment that are not climate-driven (e.g. land clearing), or could just vary internally.

Page 12, line 23: ...this lake microfossils... – should 'this' be 'thus'?

Page 12, line 31: ... analysed lead waxes... – should 'lead' be 'leaf'?

Page 13, line 5: Does not make sense – The Lake Keilambete grain-size derived is very commonly used for validation and comparison of palaeocliamte palaeoclimate records in the region

Page 13, line 8: ...accelerator mass spectrometry...

Page 13, line 18: ...could be influencesd by climate...

Page 13, line 19: ...lake cores are...

Page 13, line 31: ...assumed to be have been...

Page 14, lines 11-17: This paragraph does not belong here as is discusses an archive, not a proxy. It would be more suitable at the start of section 4.1.1. and the title adjusted to include peat.

Page 14, line 27: ...impacts...

Page 14, line 29: ...highly citesd records...

Page 15, line 7: Herbert

Page 16, line 9: ...this intensifying... – should 'this' be 'thus'?

Page 16, line 26: ...in seas-surface...

Page 17, line 24: records

Page 18, line 8: climate

Page 18, line 11: showsn

Page 19, line 33: ...number of chronological...

Page 20, line 20: ...demonstrate that that this criterion...

Page 20, line 10: . . .with regards to age-depth. . .

Page 20, lines 17-18: Sentence is not necessary

Page 20, line 32: records

Page 21, line 6: . . .this providing. . . – should 'this' be 'thus'?

Page 21, line 6: Deiffenbacher-Krall et al., 2007 should be Dieffenbacher

Page 21, line 9: . . .used to reconstruct. . .

Page 21, line 25: . . .unknown complications; Cconversely, modelling. . .

Page 22, line 17: Sentence starting 'However' is not clear.

Page 22, line 19: . . .used as an anchor. . .

Page 22, lines 27-29: Provide an Australian and ideally New Zealand reference at the end of the sentence.

Page 23, line 12: . . .limited by the short length. . .

Page 23, line 26: . . .reconstructions where proxies. . .

Page 24, lines 2-4: Sentence starting 'For example' is not clear.

Page 24, line 14: . . .signals in a regional palaeoclimate. . .

Page 24, line 30: . . .vital importantce for. . .

Page 25, line 4: . . .strength of a climate. . .

Page 25, line 24: diervisty diversity

Page 25, line 25: . . .Aus2k records network.

Page 25, line 31: . . .model comparison, and. . .

Page 26, line 3: 'for Common Era research' is not necessary

Page 26, lines 6-7: Do the authors mean 'high-resolution' or 'low resolution'?

Page 26, line 7: . . .recognizsed. . .

Response: All highlighted typographic errors will be corrected, and all coauthors will contribute to careful editing of the manuscript for the sake of identifying and correcting any additional errors.

Supplementary material Figure S1: Make axes the same units and scales Figure S2: Make axes the same units and scales

Response: The axes and scales for both figure S1 and S2 will be equalised.

References There are a number of references missing from the reference list or text. Below are the ones I found. I recommend the authors check through all text, references and supplementary material to make sure all references are included and there are no typographical errors.

The following references are missing from the reference list:

Browning and Goodwin 2014

De Deckker et al. 2011

Emile-Geay and Eshleman, 2013

Gingele et al., 2007

Goring et al. 2012

Gouramanis et al. 2010

Grant 1985

Jones et al. 1998, 2001

Kershaw 1982

Martin et al. 2014

R Development core team 2013

Schaefer et al., 2009

Response: Each in-text reference will be checked to ensure that it appears correctly in the reference list.

In the reference list:

Typo: Bowler, J. M. & Hamada, T. 1971. Late Quaternary stratigraphy and radiocarbon chronology of water level fluctuations in Lake Keilambete, Victoria. Nature, 232, 330-&. Please recheck the references for the correct format of surnames. Below are the ones I noticed.

D'costa should be D'Costa

McTainsh (Hesse, P. P. & Mctainsh, G. H. 2003. Australian dust deposits: modern processes and the Quaternary record. Quaternary Science Reviews, 22, 2007-2035)

Mcdonald should be McDonald

Mcfadgen should be McFadgen

Mcglone should be McGlone

Mckay should be McKay

McMillan (Fairchild, I. J. & Mcmillan, E. A. 2007. Speleothems as indicators of wet and dry periods. International Journal of Speleology, 36, 69-74)

LeGrande (Lewis, S. C. & Legrande, A. N. 2015. Stability of ENSO and its tropical Pacific teleconnections over the Last Millennium. Climate of the Past, 11, 1347-1360)

Response: The format of references will be checked and brought into line with the journal style requirements.

---

## Author Comment (AC4) · 23 Jun 2017

General comments

This is a very clearly written manuscript, making a convincing case for compiling multiple low-resolution archives of past environmental/climate change in the Australasian region. The paper could be useful for future palaeo-studies in the region and could inspire research teams to produce similar compilations for other regions.

Specific comments

Which calibration curve was used for the terrestrial sites, SHCal13? Make this clear within the methods.

[Figure]

Response: SHCal13 was used for terrestrial sites, and Marine13 was used for marine sites. This selection will be clarified in the methods section.

For the marine sites, how were marine dR values and their uncertainties estimated, e.g. using http://calib.org/marine/ ? Which data-points were used to estimate dR values for each site? Please provide this information as supplementary information or at your NOAA archive, so that others can replicate your findings.

Response: Marine dR values and uncertainties were taken from the original publications. This will be clarified in the methods section. An additional column in table 1 will provide links to the NOAA archive of the original publication, so that others can easily find the original dR values.

p1 line 16, what are progressive Bayesian techniques?

Response: The word 'progressive' will be removed for improved clarity of the abstract.

p2 line 4, but one could argue that during this recent time, human impact might have affected more of the proxy records. Could this potentially be a problem in some of your sites?

Response: Yes, it is possible that human impact has affected the proxy records during this time. This point will be clarified in the introduction of the manuscript. However, there is potential for human impact at longer time scales, particularly in regions of the world with long occupation histories. The authors argue that the chronological constraints available for palaeoclimate records during the last 2000 years provide a vital opportunity to investigate potential human impacts at individual sites as well as investigation of climate signals. Within New Zealand, human impact is only recognised during the last millennium (Horrocks et al., 2007, McGlone and Wilmshurst, 1999)( Horrocks, M., Nichol, S. L., Augustinus, P. C. & Barber, I. G. 2007. Late Quaternary environments, vegetation and agriculture in northern New Zealand. Journal of Quaternary Science, 22, 267-279.; Mcglone, M. S. & Wilmshurst, J. M. 1999. Dating initial Maori environmental impact in New Zealand. Quaternary International, 59, 5-16.). For this reason, comparison of records from the first millennium CE versus the second millennium CE could highlight potential human impact on palaeoclimate proxies.One of the selection criteria for high-quality records in this study is 'a demonstrated relationship between the proxy(ies) and at least one climate variable, as stated in a peer reviewed publication'. It is the assumption of this study that this criterion will identify the records where the climate signal is stronger than any potential human impact. In many records, the climate signal and human impact can be independently identified through a multi-proxy approach and/or lab-based theoretical investigations of proxy-climate relationships. The high quality records are available in their entirety on the NOAA data center. Individual researchers may make the choice to exclude the most recent section of any given record, which is the time period most likely to have human impacts.

p6 line 12, don't forget to list the error associated with having non-dated levels, and thus requiring an age-model that provides realistic estimates of uncertainties (as you explain later, on p7 lines 16-24). Perhaps cite Bennett, K.D. 1994 (The Holocene 4, 337-348), Telford et al. 2004 (Quat. Sci. Rev. 23, 1-5), and Trachel & Telford 2016 (The Holocene doi:10.1177/0959683616675939).

Response: A sentence discussing the age uncertainties in sedimentary records, as contained within undated layers, and the need for age modelling to estimate interpolation uncertainty will be added. The suggested references will be included.

Perhaps cite Flantua et al 2016 (Climate of the Past 12, 387-414) for another recent compilation of regional chronologies.

Response: Although there is some similarity between the approaches, the authors feel that Flantua et al., (2016) does not support any of the specific points made in this paper. Future work could discuss the similarities and differences between the PAGES2k regions, but such a comparison is outside the scope of this paper.

p8, line 18, Bacon does not exclude outliers but deals with them through using studentt distributions for all dates as default (not student-t tests) - these distributions look much like normal distributions but have wider tails. As a result even dates that seem outlying to our eyes (i.e. lying far away from the model and neighbouring dates) will often still fit the age-model (probability distribution >0 at the age-model at the depth of said date).

Response: The authors acknowledge the incorrect and unclear information concerning the identification and treatment of outliers within the BACON package. This section will be updated and clarified for more correct information about outliers in this study.

Language

p12 lines 22-24, check sentence

p13 line 12, associated & line 18, influenced

p17 line 29, Indonesian

p18 line 11, shown

p22 line 4, renewed efforts to renewed efforts to

p25 line 24, diversity

p26 line 25, Past Global Changes (not glocal)

Response: The authors acknowledge the typos identified by reviewer #2. The manuscript will be carefully edited by all coauthors in order to identify and correct these, and any additional, typographic errors.

---

## Author Response (AR1)

[revised manuscript text omitted]

*Pollen*

Australasia has a rich history of palynological studies, with research extending back to the 1960s (Churchill, 1960; Moar, 1967). Early research focused on reconstructing vegetation diversity at a single site, but approaches have expanded to examine broader environmental questions such as regional vegetation response to climatic shifts (Donders et al., 2007), reconstruction of a specific variable across a region (Fletcher and Thomas, 2010a), recovery from episodes of disturbance such as fire (Lynch et al., 2007), and responses to human impacts in the pollen catchment (Haberle et al., 2006; Horrocks et al., 2001; Leahy et al., 2005).

The investigation of the abundance and ecological assemblage of pollen spores sheds lights on palaeoecological dynamics through time, predominantly driven by changes in climate (Donders et al., 2007; Kershaw et al., 1991) and the impacts by human activity (Lynch et al., 2007). There is an extensive network of well-studied sites centred on the Atherton Tablelands in northern Queensland (Haberle, 2005; Haberle et al., 2006; Kershaw, 1970, 1975, 1983, 1971; Kershaw et al., 2007; Walker et al., 2000), including some of the most highly cited records of Australian-Indonesian Summer Monsoon dynamics and rainforest response to climate variability through time (Kershaw, 1994). The majority of pollen records are restricted to the peripheries of the Australian continent due to the need for water availability in the accumulation and preservation of pollen grains (Fitzsimmons et al., 2013). There have been comparisons of available pollen records for southeast Australia (D'Costa and Kershaw, 1997), and a north-south transect of high-quality pollen records along the east coast of Australia (Donders et al., 2007). New Zealand palynology records cover the entire length of the country, and range in resolution from millennial scale to multi-decadal scale (see references in Lorrey and Bostock, 2017).

While almost all Australasian pollen reconstructions are qualitative in their approach, there have been a small number of quantitative studies. Cook and Van der Kaars (2006) comprehensively outlined early approaches (i.e. single-taxa indicators and modern analogue techniques (MAT)) and their limitations in the Australian context. The same study explored the potential of existing pollen sites to be used for construction of transfer functions, and found that regional transfer functions could be used to associate modern pollen distributions to modern hydroclimate. Herbert and Harrison (2016) conducted a similar review of modern analogue techniques in Australia and suggested that, despite possible limitations in the current sampling density of the continent, MAT can be an appropriate reconstruction technique. Transfer functions have been produced for average annual temperature in Tasmania (Fletcher and Thomas, 2010a) and New Zealand (Wilmshurst et al., 2007), but no quantitative reconstructions that pass the Aus2k criteria encompass the Common Era. Development of training datasets in New Zealand has historically been complicated by deforestation since 750YBP; however, a pre-deforestation dataset developed by Wilmshurst et al. (2007) mitigates this issue for future New Zealand studies.
* * *
**Bronwyn Christina Dixon 6/7/17 2:43 PM**
**Moved up [4]:** Peat bogs can be ideal sources of sediment cores because of high accumulation rates and the capture of a variety of organic and non-organic components (Barbar et al., 1994, Booth et al., 2010). Peat is a promising archive within Australasia due to the large longitudinal coverage; mires are found from tropical locations in Papua New Guinea to the sub-Antarctic islands (McGlone 2002b, McGlone et al., 2010, Whinam and Hope, 2005). Most peat-based records in Australasia contain pollen and charcoal that may be used for palaeoclimate, palaeoecology, and paleo-fire reconstructions (Whinam and Hope 2005). Degree of humification has also been applied as a hydroclimate indicator in the late Holocene (Burrows et al., 2014, Wilmshurst et al., 2002) in cores were dating density is sufficient.

**Bronwyn Christina Di…, 28/7/17 11:01 AM**

**Bronwyn Christina Dixon 6/7/17 2:48 PM**

**Bronwyn Christina Dixon 6/7/17 2:49 PM**

**Bronwyn Christina Dixon 5/7/17 3:31 PM**
**Moved up [3]:** The primary consideration in lake, wetland, or peat-derived climate archives is a clear understanding of the modern proxy~climate relationship, including the geomorphic, hydrological, geochemical, and biological response, most of which are complex and non-linear. Detailed understanding of these relationships relies on local monitoring programs or comparisons between observed proxy behaviour and instrumental records over a sufficient length of time. Development of mechanistic response models is also a priority, in order to quantify the sensitivity of certain proxies to hypothetical changes, as well as to integrate the multiple effects upon a particular palaeoclimate signal.
The most common approach in Australia is the 'calibration in space' of lake and wetland derived proxies, where lakes across an environmental gradient are sampled for the purpose of establishing modern proxy-limnology-climate relationships (Gell, 1997, Saunders, 2011). Comparisons with instrumental records form the basis of the 'calibration in time' technique, where a proxy time series is calibrated with meteorological data to produce a predictive function for quantitative reconstruction for a longer time series (Larocque-Tobler et al., 2011). Two studies within the Aus2k databset have used 20th century instrumental precipitation and temperature data for calibration of longer records of sediment reflection data (Saunders et al., 2013, Saunders et al., 2012). A possible limitation in the Australasian context is the ... [8]

**4.1.2 Marine cores**

[revised manuscript text omitted]

**Response to Reviewer 1**

**General comments**

A huge effort would have gone into compiling these records and the manuscript represents an important contribution to the PAGES2K network and the palaeoclimate community more generally.

My main comments relate to providing some additional information so readers can more easily assess old vs. new age-depth models and reproduce them, together with more of a critique about causes of differences and factors to be aware of. At present this is not possible, which detracts from the value of the paper. It would also be helpful to include direct links to each record that has been archived (see 2 below).

**Response**:

This identification and assessment of non-annually resolved Australasian palaeoclimate records has been underway since 2009, and has been the focus of both post-doctorate research and a PhD project. We are pleased to present the outcomes of more than eight years of work, and hope that the scientific community can benefit. Original and updated age models and datasets will be available through NOAA World Data Center at:

ftp://ftp.ncdc.noaa.gov/pub/data/paleo/pages2k/dixon2017australasia/

This link will become publicly available upon publication of this manuscript.

**Specific comments**

**Abstract**

* Include briefly: where the best and worst coverage of sites is; the main reason(s) for the differences between old and new age-depth models; and summary of recommendations.

**Response:** The abstract was revised as suggested and now reads as follows:

'Non-annually resolved palaeoclimate records in the Australasian region were compiled to facilitate investigations of decadal to centennial climate variability

over the past 2000 years. A total of 675 lake/wetland, geomorphic, marine, and speleothem records were identified. The majority of records are located near population centres in southeast Australia, in New Zealand, and across the maritime continent, and there are few records from the arid regions of central and western Australia. Each record was assessed against a set of *a priori* criteria based on temporal resolution, record length, dating methods, and confidence in the proxy-climate relationship over the Common Era. A subset of 22 records met the criteria, and was endorsed for subsequent analyses. Chronological uncertainty was the primary reason why records did not meet the selection criteria. New chronologies based on Bayesian techniques were constructed for the high quality subset to ensure a consistent approach to age modelling and quantification of age uncertainties. The primary reasons for differences between published and reconstructed age-depth models were the consideration of the non-singular distribution of ages in calibrated $^{14}C$ dates and the use of estimated autocorrelation between sampled depths as a constraint for changes in accumulation rate. Existing proxies and reconstruction techniques that successfully capture climate variability in the region show potential to address spatial gaps and expand the range of climate variables covering the last 2000 years in the Australasian region. Future palaeoclimate research and records in Australasia could be greatly improved through three main actions: i.) Greater data availability through the public archiving of published records, ii.) Thorough characterisation of proxy-climate relationships through site monitoring and climate sensitivity tests, and iii.) Improvement of chronologies through core-top dating, inclusion of tephra layers where possible, and increased date density during the Common Era.'

**2 Data and Methods**

* As it is stated that many of the records were not publically available, were some obtained by personal communication with the original authors? If so, this should be noted in addition to NOAA and Neotoma databases.

**Response:** Sources of pre-existing records have been updated to include 'a general inquiry to Australasian Quaternary Association members' and 'personal communication with authors'

* Can links to the NOAA database for each dataset in Table 1 be included? This means readers can go directly to the relevant record.

**Response:** Links to the NOAA URL for originally published datasets are now included in Table 1. A link to the comparison of originally published and updated age models are also included in the 'Data availability' section at the end of the manuscript.

**3.3 Age model updates and 4.2 Discussion of age modelling approaches**

* It would be helpful if the BACON settings for each model are provided (e.g. in supplementary material). This would ensure people can reproduce the agedepth models exactly. For example, values specified for thickness, accumulation mean and memory strength influence the output, and these are determined based on prior knowledge of the core and site as the authors highlight on page 8. If this information is not included others will not necessarily be able to reproduce the chronologies.

**Response:** A supplement of all code and settings used to construct updated age models are now included in the supplementary information, for the purpose of transparency and reproducibility of methods presented in this study

* In Figure 4, can the original age-depth models be plotted on top of the new BACON-derived outputs (or at least side by side as done in Figure S2) so readers can see the differences between old and new? At present it is not possible to make this assessment. It would also be helpful if examples were provided of cores where new and old age-depth models are quite similar, and some examples where there are notable differences, together with a critique as to why.

**Response:** Figure 4 has been updated to show the four example age-depth models side-by-side with the originally published age models so that differences can be observed. Each pair of old/new age models has been provided in the supplementary information, so that readers can see examples of similar and divergent age models. Further discussion of reasons for differences in age model outcomes has been included within section 4.2, which now reads: 'Chronologies for most marine sediment cores are based on fitting a linear age-depth relationship across the set of dates. The results of BACON-derived age models moderately support the application of linear accumulation for this archive, with the ages of young/shallow samples more likely to match between the two approaches in comparison to samples from deeper in the core. The disparity between published and reconstructed ages for marine records increases back through time. This is most likely related to date density through individual cores, as well as periods of interpolation between dated horizons. Indonesian marine cores use the 1815 Tambora tephra as a chronological anchor, which decreases age uncertainty near the present, but date density further back in time varies between records.

The published Snowy Mountain core had age model difficulty, with two possible age models with similar $r^2$ values (Marx et al., 2011). The self-adjusting Monte-Carlo approach within the BACON software clearly favoured one model over the other (Supplementary Fig. 1). The BACON-derived age model showed greater agreement with the published age model built upon more dates. This is likely driven by acknowledgement of the probability distributions of calibrated radiocarbon dates, as well as estimated autoregression between sampled (but undated) depths.'

* While the authors state that the decisions by the original authors regarding exclusion of radiocarbon dates were upheld, was this also the case for the Lake Elingamite record (Figure S2)? The CLAM-model shows three dates (in red) that do not appear to be in the BACON model output. If these were treated as outliers

by Barr et al. (2014) but still plotted, which is an option in CLAM, this should be noted in the figure caption to explain why the number of dates is different.

**Response:** The dates presented in red in Barr et al., (2014) have indeed been excluded by this study because of their previous identification as outliers. Figure S2 has been updated to compare original and BACON age models for all records in the Aus2k dataset.

**4.1.1 Lakes and wetlands.**

* The authors list the common factors that can have an impact on preservation of the climate signal (e.g. sediment accumulation rate, basin morphology, page 11, line 4). Human impacts, particularly since European settlement can also override potential climate signals. This can complicate the development of transfer functions/modern analogue technique models and calibration in time. This is highlighted in the 'lithics' section, but is relevant to biological proxies too.

**Response:** The information on biological proxies in section 4.1.1. has updated to appropriately communicate the impact of site-specific feature and land use change on biological proxies in Australasia. Relevant supporting literation has also been included.

* Page 14, lines 8-10 and Supplementary Fig. 1: State which age model the BACON-derived one supports and why.

**Response:** This section has clarified and expanded upon to indicate which published model is preferred, as indicated by the BACON-derived age model, and why BACON is a useful tool for choosing appropriate age models. It now reads: 'The published Snowy Mountain core had age model difficulty, with two possible age models with similar $r^2$ values (Marx et al., 2011). The self-adjusting Monte-Carlo approach within the BACON software clearly favoured one model over the other (Supplementary Fig. 1). The BACON-derived age model showed greater agreement with the published age model built upon more dates. This is likely driven by acknowledgement of the probability distributions of calibrated radiocarbon dates, as well as estimated autoregression between sampled (but undated) depths.'

* Page 15, lines 14-19: This paragraph is better suited near the start of the section before discussing the different proxies as it is general. Some references would be beneficial, in particular in relation to proxies being complex and non-linear. In addition, using a multi-proxy approach is important for being able to potentially discern climate vs. human impact vs. within lake signals.

**Response:** This paragraph has shifted to the beginning of the section (page 11, lines 23-33), and will be edited to prevent repetition with the existing introductory paragraph. Supporting literature has been added to support the complex and non-linear nature of palaeoclimate proxies in lake and wetland environments.

* Page 15, line 24: von Gunten et al. (2012) is an additional reference as it describes the calibration in time approach using case studies based on biogeochemistry (von Gunten L, Grosjean M, Kamenik C, Fujak M, Urrutia R (2012) Calibrating biogeochemical and physical climate proxies from non-varved lake sediments with meteorological data: methods and case studies. J Paleolimnol DOI 10.1007/s10933-012-9582-9).

**Response:** Information from this reference has been incorporated into page 12, line 6-8, and the paper has been added to the reference list.

*Page 19, lines 4-7: the difference between the oldest/closest to 1 CE dates are also likely due to the amount of extrapolation between it and the previous 14C date, not just the density of dates through the core. In addition to having at least one date near the top of the core (line 8), ideally 210Pb and/or 137Cs would be used if the sedimentation rate is fast enough.

**Response:** The degree of interpolation will be discussed in the possible reasons for large chronological uncertainties. References will be supplied to support this point. The authors agree that $^{210}$Pb and/or $^{137}$Cs would be ideal for constraining the top of sediment cores in Australasia, and these radionuclides have has been used in previous studies. The use of 210Pb dating for constraining upper-core ages is included in the 'chronology' section of the discussion (page 23, lines 4-9).

**Figures and Tables**

**\* Figures 1 and 3:** What symbol marks peat records?
**Response:** Peat records are included in the lake/wetland category, which is represented by the purple markers. The figure captions have been changed to clarify the types of proxies included within each archive category.

**\* Tables:** It is not clear how studies are ordered, which makes it hard to search through them (also applies to Table S1)

**Response:** Records within all tables in the main text and supplementary section have been reordered by archive type (i.e. lake/wetland, marine, speleothem, marine), then by state/country, for the sake of easy searching by readers.

**\* Table 2**
Just presenting the difference between top and base years in the old compared to new chronologies is useful, but does not necessarily illustrate the actual differences between the age-depth models. For example, if the lowermost 14C date is above the bottom of the core, which means the age-depth models are extrapolated, this may lead to a larger apparent difference in ages than might be the case for most of the core. To address this, figures of each site with old and new age-depth models could be included. All new age-depth model figures should already be available as part of BACON output. Doing this means the settings could be incorporated into each figure (see comment 3.3 above). Ideally the original age-depth models would be plotted on top to best illustrate

differences and similarities. If this is not possible, then at least provide them side-by-side. This would help readers assess the differences for themselves, identify common patterns and assist their decision making when investigating these records and developing chronologies for other sites.

**Response:** Comparison figures displaying overlain old and new age models for each of the 22 sites are now included in the supplementary material in Supp Fig. 2. Table 2 has been removed from the main text and readers now directed to the supplementary Table S2.

**\* Technical corrections:** There are a number of typographical errors in the text. I have listed the ones I found, but recommend the authors do a thorough check. This includes the order of references within the text, which are not always consistent (e.g. Marx et al., 2011, Marx et al., 2009, page 13, line 32).

\*Page 1, lines 15-16: A high quality subset of 22 records across Australasia met the criteria and they are **were** endorsed for subsequent analyses

**Response:** Revised to read: 'A subset of 22 records met the criteria, and was endorsed for subsequent analyses.'

\*Page 2, line 30: Low-resolution sedimentary archived **archives** available within Australasia include lacustrine...

**Response:** The text now reads: 'Low-resolution sedimentary archives available within Australasia include lacustrine, fluvial and wetland sediments (including peat), marine sediments, speleothems, and geomorphic features (e.g. moraines, dunes) across diverse climate settings.' (Page 3, lines 4-6)

\* Page 3, line 13: state the most recent year of publication of the records so it is clear until when the database is up to date. This is important because new records are being published (e.g. comments by Rouillard).

**Response:** The text now reads 'Records published before 2017 were identified through inspection of citation databases, reference lists of past review papers, online public data repositories, personal communication with authors, and a general inquiry to all Australasian Quaternary Association members.' (Page 4, lines 2-4)

\* Page 4, line 12: a reference is needed at the end of the sentence The Australasian region includes tropical Southeast Asia because of the dynamical influences of the Indo-Pacific region on the Australasian monsoon. As two Antarctic sites appear to be included, the reason why stated.

**Response:** References have been added to this section, and the text now reads: 'The Australasian region includes tropical Southeast Asia because of the dynamical influences of the Indo-Pacific region on the Australasian monsoon (Meehl and Arblaster, 2002), as well as the Australian/New Zealand sector of the Southern Ocean and Antarctica because of oceanographic and atmospheric teleconnections

(Hall and Visbeck, 2002; van Ommen and Morgan, 2010).' (Page 4, lines 17-20)

Page 4, line 19: 'Reasonable' was defined as **by** PAGES2k as containing at least one…

**Response:** The correction has been made, and the text now reads: ''Reasonable' was defined by PAGES2k as containing at least one chronological control point near the youngest part of the record, another near 1CE or the end of the record (whichever is younger), and, for records greater than 1000 years in length, an additional date near the middle of the record.' (Page 4, lines 27-29)

* Page 5, line 1: …approach for the creation of age models in presented in this study…

**Response:** This sentence has been clarified, and the text now reads: 'Although one approach for the creation of age models is presented in this study, the raw data are available for individuals who wish to apply alternative methods.' (Page 5, lines 9-11).

* Page 5, line 8: Or **Of** these 241 records…

**Response:** The correction has been made, and the text now reads: 'Of these 241 records, 141 were classified as 'moderate to high confidence' based on climatic sensitivity, possible non-climatic influences, local forcing, and chronological confidence.' (Page 5, lines 17-18)

* Page 5, line 27: Should temperature be temperate?

**Response:** This correction has been made, and the text now reads: 'These data were compiled from a site level up to a homogeneous regional climate district level (Kidson, 2000; Lorrey et al., 2007) from a range of environments that range from temperate subtropical in the far north of New Zealand to glacial in the south.' (Page 6, lines 1-3)

Page 6, line 27: …residence time within a radiocarbon samples…

**Response:** The correction has been made, and the text now reads: 'It is not always possible to identify environmental residence time within a radiocarbon sample, and the uncertainty may not be acknowledged within the resulting chronology (McFadgen, 2007).' (Page 7, lines 4-5)

* Page 7, lines 26-27: Sentence is not necessary – In this study, one focus is to generate new age models for records that meet the PAGES2k selection criteria, providing consistency in the approach to age determination and uncertainty estimates.

**Response:** This sentence has been removed from the manuscript.

* Page 7, lines 27-29: Sentence is not necessary as it overlaps with the end of the previous paragraph. Combine it with the previous paragraph so the references are included. This study applies Bayesian age modelling across the Aus2k records, a decision that follows the initiative of the wider palaeoclimate community (e.g. Anchukaitis and Tierney, 2012, Goring et al., 2012, Hua et al., 2012).

**Response:** This sentence has been removed, and the references were added to the previous paragraph.

* Page 9, lines 13-14: Sentence not necessary – 661 Australasian sedimentary records spanning the Common Era were systematically reviewed for their suitability for reconstructing regional climate dynamics over the last 2000 years.

**Response:** Sentence was removed.

* Page 9, lines 21-22: Sentence is a repeat of the previous section – Lacustrine microfossils are the most common terrestrial proxy in the Aus2k records, while foraminifera geochemistry is the predominant marine proxy.

**Response:** This sentence discusses the composition of the vetted Aus2k dataset, rather than the complete regional dataset.

* Page 10, line 5: Missing word: A low number **of** radiocarbon dates…

**Response:** The missing word was added to the sentence. The text now reads: 'A low number of radiocarbon dates and/or low confidence in the chronologies are the most common reasons for record exclusion from the Aus2k dataset.' (Page 10, lines 12-14)

* Page 10, line 11: …resolution to intestigate **investigate** decadal…

**Response:** The typo was corrected, and the text now reads: 'The 22 records that meet the PAGES2k selection criteria provide a subset of records with robust chronologies, an identified proxy-climate relationship, and sufficient record length and resolution to investigate decadal to centennial variability during the last 2000 years.' (Page 10, lines 17-19)

* Page 10, line 15 paragraph: This appears to be a contradiction to the start of the Discussion where the authors comment there is widespread spatial and temporal coverage of existing records across the geographic network (page 10, line 1). Please reword to clarify what is meant.

**Response:** This section discusses the vetted Aus2k dataset, rather than the regional dataset. This has been clarified in the text, which now reads: "The geographic distribution of Aus2k records displays stronger spatial biases than the complete regional database (Fig. 3).' (Page10, lines 24-25)

* Page 10, lines 18-19: …climate drivers including **the** El Niño-Southern

Oscillation (ENSO), the Indian Ocean Dipole (IOD), and the Australian-Indonesian Summer Monsoon...

**Response:** The sentence has been revised, and the text now reads: 'In contrast, the availability of high quality records in Indonesia relates to global interest in the region as a dynamical 'centre of action' for numerous climate drivers including the El Niño-Southern Oscillation (ENSO), the Indian Ocean Dipole (IOD), and the Australian-Indonesian Summer Monsoon (AISM), resulting in high levels of international funding for research in this area.' (Page 10, lines 26-29)

*Page 10, line 32: Climate **Common** Era

**Response:** The text was corrected and now reads: 'A brief description of archives and their applicability to studying climate variability during the Common Era is provided below.' (Page 11, lines 9-10)

* Page 11, line 2: Core**s**

**Response:** The correct form of the word has been added to the text.

* Page 11, line 23: Chivas et al., 1985, **Chivas et al., 1985**

**Response:** The reference has been corrected in the text and in the reference list.

* Page 11, line 25: Oxygen isotopes values...

**Response:** The text has been corrected and now reads: '. Oxygen isotope values reflect the combined influence of the oxygen isotopic composition and temperature of the lake water, while carbon isotopes reflect the isotopic composition of the dissolved inorganic carbon present in the lake system (see Gouramanis et al., 2010 and references therein).' (Page 12, lines 26-28)

* Page 12, line 5: Transfer functions built upon **developed from** modern calibration...

**Response:** The text has been clarified and now reads: 'Transfer functions, developed from modern calibration data sets, have been established for numerous aquatic variables in both estuarine (Logan and Taffs, 2013; Tibby and Taffs, 2011) and lacustrine (Gell et al., 2005) settings.' (Page 13, lines 8-9)

* Page 12, line 20: ... because of the their...

**Response:** The text has been corrected and now reads: 'Over the past half-century, speleothems have emerged as valuable sources of palaeoclimate information because of their potential for preserving precisely dated, multi-proxy, high-resolution records of past climate change (Fairchild et al., 2006).' (Page 17, line 2-3)

* Page 12, line 22: ...both with on biological and statistical grounds...

**Response:** This sentence has been removed to improve the clarity of this section.

Page 12, lines 22-23: this sentence is misleading (Nevertheless, the ecological dynamics of lakes are clearly governed by climate conditions) because the ecological responses in some lakes are influenced more by humans than climate (e.g. damming), or changes in the catchment that are not climate-driven (e.g. land clearing), or could just vary internally.

**Response:** These two sentences have been removed. The consideration of the limnology-proxy-climate connection is now discussed in section 4.1.1.

Page 12, line 23: ...this lake microfossils... – should 'this' be 'thus'?

**Response:** This sentence has been removed to improve the clarity of this section.

Page 12, line 31: ... analysed lead waxes... – should 'lead' be 'leaf'?

**Response:** The text has been corrected and now reads: 'Tierney et al. (2010) analysed leaf waxes in material that had been transported offshore.' (Page 13, line 26)

Page 13, line 5: Does not make sense – The Lake Keilambete grain-size derived is very commonly used for validation and comparison of palaeocliamte **palaeoclimate** records in the region

**Response:** This sentence has been clarified and corrected, and the text now reads: 'The Lake Keilambete lake level reconstruction is very commonly used for validation of other palaeoclimate records in the region.' (Page 13, lines 33-34)

Page 13, line 8: ...accelerator mas**s** spectrometry...

**Response:** The typo has been corrected, and the text now reads: 'The dating density is also the highest of any of the Aus2k records, with four new accelerator mass spectrometry (AMS) radiocarbon dates and four optically stimulated luminescence (OSL) dates within the last 2000 years in the most recent chronology (Wilkins et al., 2012).' (Page 14, lines 1-3)

Page 13, line 18: ...could be influences**d** by climate...

**Response:** The correct word has been substituted here, and the text now reads: 'Signatures of acute sedimentation are interpreted as rapid in-washing of coarse loads (Lake Rotonuiaha (Wilmshurst et al., 1997); Lake Tutira (Eden and Page, 1998; Gomez et al., 2012); Round Lake (Chester and Prior, 2004)) that could be influenced by climate and/or seismic variability.' (Page 14, lines 10-12)

Page 13, line 19: …lake core**s** are…

**Response:** The typo has been corrected, and the text now reads: 'Some of the lithological changes observed in lake cores are suggested as a response to climatically driven increases and decreases in lake level (Lake Maratoto (Green and Lowe, 1985); Lake Poukawa (McGlone, 2002b)).' (Page 14, lines 12-14)

Page 13, line 31: …assumed to be have been…

**Response:** This sentence has been clarified and now reads: 'For coastal study sites, the local quartz input from sand dunes is ignored, while remaining material is assumed to have been transported to the site via aeolian means (McGowan et al., 2008).' (Page 14, lines 24-25)

Page 14, lines 11-17: This paragraph does not belong here as is discusses an archive, not a proxy. It would be more suitable at the start of section 4.1.1. and the title adjusted to include peat.

**Response:** This paragraph has been added to the beginning of section 4.1.1.

Page 14, line 27: …impact**s**…

**Response:** The text has been corrected, and now reads: 'Early research focused on reconstructing vegetation diversity at a single site, but approaches have expanded to examine broader environmental questions such as regional vegetation response to climatic shifts (Donders et al., 2007), reconstruction of a specific variable across a region (Fletcher and Thomas, 2010a), recovery from episodes of disturbance such as fire (Lynch et al., 2007), and responses to human impacts in the pollen catchment (Haberle et al., 2006; Horrocks et al., 2001; Leahy et al., 2005).' (Page 15, line 5-9)

Page 14, line 29: …highly cites**d** records…

**Response:** The typo has been corrected, and the text now reads: 'There is an extensive network of well-studied sites centered on the Atherton Tablelands in northern Queensland (Haberle, 2005; Haberle et al., 2006; Kershaw, 1970, 1975, 1983, 1971; Kershaw et al., 2007; Walker et al., 2000), including some of the most highly cited records of Australian-Indonesian Summer Monsoon dynamics and rainforest response to climate variability through time (Kershaw, 1994).' (Page 15, lines 12-15)

Page 15, line 7: Herber**t**

**Response:** The typo has been corrected (Page 15, line 25)

Page 16, line 9: …this intensifying… – should 'this' be 'thus'?

**Response:** The typo has been corrected and the text now reads: 'For some coupled ocean-atmosphere climate modes, such as ENSO and the IOD, the coupled

warm/wet and cold/dry conditions influence the $\delta^{18}O$ signature in the same direction, thus intensifying the climate signal in the oxygen isotopes (Brijker et al., 2007; Khider et al., 2011)' (Page 16, lines 11-13)

Page 16, line 26: ...in seas-surface...

**Response:** The typo has been corrected, and the text now reads: '. It is also important to separate the atmospheric and oceanographic drivers of variability in sea-surface temperatures and salinity through multi-proxy approaches.' (Page 16, line 29-30)

Page 17, line 24: records

**Response:** The typo has been corrected and the text now reads: 'Early- to mid-Holocene $\delta^{13}C$ in a Tasmanian speleothem record was inferred to reflect productivity and the resulting carbon isotope fractionation in the soil (Xia et al., 2001).' (Page 17, lines 31-32)

Page 18, line 8: cli**m**ate

**Response:** The typo has been corrected and the text now reads: 'This synoptic approach allows the integration of co-varying, spatially heterogeneous responses of several speleothem environments to surface climate, which is forced by orographic circulation and advection related to base climate state shifts (Lorrey et al., 2008; Lorrey et al., 2014; Lorrey et al., 2012).' (Page 18, lines 14-17)

Page 18, line 11: shows**n**

**Response:** The typo has been corrected and the text now reads: 'In the Australian setting, Mg/Ca in speleothems has been shown to be a reliable recorder of effective rainfall (Fairchild and Treble, 2009; Treble et al., 2003; McDonald et al., 2004) because longer water residence times increase the Mg/Ca in speleothem drip water (Fairchild et al., 2000; Fairchild and McMillan, 2007).' (Page 18, lines 18-20)

Page 19, line 33: ...number **of** chronological...

**Response:** The word has been added, and the text now reads: 'The records that have the greatest departure from the published age models are those that have the smallest number of chronological anchors.' (Page 20 lines 11-12)

Page 20, line 20: ...demonstrate that that **this** criterion...

**Response:** The typo has been corrected and the text now reads: 'However, the outcomes of this study demonstrate that this criterion is still too relaxed for robust investigation of decadal to multi-decadal climate variability in the Common Era.' (Page 20, lines 14-15)

Page 20, line 10: ...with regards **to** age-depth...

**Response:** The sentence was corrected and the text now reads: 'It can be argued that the Bayesian methods applied in this study represent the current state of the art with regards to age-depth modelling of Quaternary sediment sequences.' (Page 20, line 23-24)

Page 20, lines 17-18: Sentence is not necessary

**Response:** This sentence has been removed for the clarity of the section.

Page 20, line 32: record**s**

**Response:** The typo has been corrected. The text now reads 'Only one low-resolution terrestrial temperature reconstruction exists in the Aus2k dataset: the Duckhole Lake record (Saunders et al., 2013).' (Page 21, lines 10-11)

Page 21, line 6: …this providing… – should 'this' be 'thus'?

**Response:** The typo has been corrected and the text now reads: 'All of these transfer functions have the potential of being applied to the Common Era, thus providing quantitative reconstructions from sub-tropical and temperature climate zones of Australasia.' (Page 21, lines 17-18)

Page 21, line 6: D**ei**ffenbacher-Krall et al., 2007 should be D**ie**ffenbacher

**Response:** The reference typo has been corrected.

Page 21, line 9: …used to **re**construct…

**Response:** The typo has been corrected and the text now reads: 'This proxy has been used to reconstruct mean annual temperature in Lake Pupuke in New Zealand (Heyng et al., 2015) and Lake Mackenzie in Australia (Woltering et al., 2014).' (Page 21, lines 20-22)

Page 21, line 25: …unknown complications; C**c**onversely, modelling…

**Response:** The capitalisation has been removed (Page 22, line 3)

Page 22, line 17: Sentence starting 'However' is not clear.

**Response:** The sentence has been clarified and the text now reads: 'There are now low-cost commercial services, and institutional dating operations have endeavoured to match the costs of the competitive commercial market. These developments mean new, more cost-effective dating strategies can be employed to create new well-dated records, as well as renewed efforts to revisit former sites to improve the original chronologies.' (Page 22, lines 12-15)

Page 22, line 19: …used **as** an anchor…

**Response:** The text has been corrected and now reads: 'The only eruption used as an anchor in the Aus2k dataset is Tambora (1815CE) in the Malay Archipelago records.' (Page 22, lines 29-30)

Page 22, lines 27-29: Provide an Australian and ideally New Zealand reference at the end of the sentence.

**Response:** Australian and New Zealand references have been added and the text now reads: 'When possible, conducting core-top radionuclide analyses, such as $^{210}$Pb and $^{137}$Cs (Appleby and Oldfield, 1978) can offer greater confidence in the age at the top of the core, as well as any significant impacts on the site by the arrival of Europeans (Sloss et al., 2011; Rodysill et al., 2013; Roop et al., 2016).' (Page 23, lines 4-6)

Page 23, line 12: ...limited by **the** short length...

**Response:** The sentence has been corrected and the text now reads: 'All of the approaches are potentially limited by the short length and quality of calibration time scales, uncertainties in proxy archive dating, regional biases from uneven spatial coverage, seasonal sensitivity, and in some cases multiple influences on proxy archive interpretation (i.e. potential distortion effects from other environmental processes).' (Page 23, lines 21-23)

Page 23, line 26: ...reconstructions w**h**ere proxies...

**Response:** The text has been clarified and now reads: 'Past climate interpretations using this approach remain heavily reliant on modern observations, palaeoclimate reconstructions that have been calibrated to local climate data, sufficient palaeodata network density, and understanding how other forcing mechanisms operated and impacted local climates in the past.' (Page 24, lines 1-4)

Page 24, lines 2-4: Sentence starting 'For example' is not clear.

**Response:** The text has been clarified and now reads: 'The use of 'upstream' sites could assist in improving the skill of reconstructions for a particular proxy at locations of interest. The utilisation of high-quality sites near to, but outside, the location of interest may lead to regional reconstructions with higher statistical skill, as they may preserve signals of large-scale circulation patterns rather than local climate features (Gallant and Gergis, 2011; Gergis et al., 2012; Gergis et al., 2016; Ho et al., 2013). In Australia, sites along the southern coast of South Australia and Victoria are impacted by a similar atmospheric circulation features and remote climate drivers as major cities and agricultural centres of southern Australia (Murphy and Timbal, 2008), lending the potential to use palaeoclimate records from near the coast to infer patterns of change inland (Ho et al., 2013).' (Page 24, lines 11-17)

Page 24, line 14: ...signals in **a** regional palaeoclimate...

**Response:** The sentence has been corrected and now reads: 'In particular, the resulting spatial fields and climate metrics (indices, archetypal patterns) that are able to be generated from the current range of approaches used by the Aus2k group have offered opportunities to explain how local signals in a regional palaeoclimate network simultaneously arise in a dynamical context.' (Page 24, lines 23-25)

Page 24, line 30: ...vital importantce for...

**Response:** The typo has been corrected and the text now reads: 'It is of vital importance for continuation of data comparison in climate research that those creating records archive their existing and future data with at least one of those repositories.' (Page 25, lines 6-8)

Page 25, line 4: ...strength of **a** climate...

**Response:** This sentence reads in the way that the authors intended. No change made.

Page 25, line 24: diervisty **diversity**

**Response:**  The typo has been corrected and the text now reads: 'Additional records of similar quality are needed to further expand the spatial coverage and diversity of climate variables within the Aus2k record network.' (Page 26, lines 7-8)

Page 25, line 25: ...Aus2k records network.

**Response:** The typo has been corrected (Page 26, lines 7-8)

Page 25, line 31: ...model comparison**,** and...

**Response:** The Oxford comma has been added to the sentence. The text now reads: 'Thorough characterisation of proxy-climate relationships could be achieved through site monitoring, climate signal characterisation through model comparison, and development and evaluation of new biological/sedimentological transfer functions.' (Page 26, lines 13-15)

Page 26, line 3: 'for Common Era research' is not necessary

**Response:** 'For Common Era research' has been removed.

Page 26, lines 6-7: Do the authors mean 'high-resolution' or 'low resolution'?

**Response:** 'High resolution' has been removed for clarity of the section. The text now reads: 'However, the existing high-quality records demonstrate the potential of sites within this region to provide well-dated records with recognised connections to climate variables.' (Page 26, lines 21-22)

Page 26, line 7: ...recogniz**s**ed...

**Response:** The American spelling has been replaced with British spelling and is now in line with the rest of the document.

**Response:** Thank you for the careful editorial reading of our work. All typographic errors were corrected as suggested

**Supplementary material**
Figure S1: Make axes the same units and scales
Figure S2: Make axes the same units and scales

Response: The axes and scales for figure S1 have been equalized.

**Response:** This sentence has been removed, and the information has been

expressed in section 4.1.1.

*p13 line 12, associated & line 18, influenced

**Response:** The words 'associated' and 'influenced' have been corrected.

* p17 line 29, Indonesian

**Response:** 'Indonesia' has been corrected to 'Indonesian'.

* p18 line 11, shown

**Response:** 'shows' has been corrected to 'shown'

* p22 line 4, renewed efforts to renewed efforts to

**Response:** Revised to read: 'These developments mean new, more cost-effective dating strategies can be employed to create new well-dated records, as well as renewed efforts to revisit former sites to improve the original chronologies.'

* p25 line 24, diversity

**Response:** The spelling of 'diversity' has been corrected.

* p26 line 25, Past Global Changes (not glocal)

**Response:** The spelling of 'Global' has been corrected.

**Response to Short Comment 1**

Dear Dr. Rouillard,

Response: The authors of "Low-resolution Australasian palaeoclimate records of the last 2000 years" thank you for bring additional publications to our attention. As you say, there is a paucity of records in western/northern Australia. For this reason, it is important to include all available records in our reference list.

The references you suggested have been added to the reference list, and the record sites within these papers will be added to our results. Unfortunately, none of the record meet all of the criteria necessary for inclusion in the 'high quality' dataset. The reason(s) for exclusion are listed after each reference.

Again, we thank you for your assistance to ensuring all records are available for future consideration by the palaeoclimate community.

Dear authors,
Considering the relative paucity of records in the Western and Northern parts of Australia, I would like bring to your attention the following recently published low resolution records for these regions, with respective references, for addition to Figure 1 & Figure 2.

Black Springs Lake Wetland McGowan et al., 2012
King River LakeWetland Proske et al., 2014
Fortescue Marsh LakeWetland Rouillard et al., 2016a, b

McGowan, H. et al. (2012) Evidence of ENSO mega-drought triggered collapse of prehistory Aboriginal society in northwest Australia. Geophysical Research Letters 39, L22702, 1–5

Response: This record is excluded from the 'high quality' record list because of an insufficient number of dates within the past 2000 years, as well as too low resolution. The PAGES selection criteria require at least three dates within the past 2000 years if the record is longer than 1000 years.

Proske, U. et al. (2014) A Holocene record of coastal landscape dynamics in the eastern Kimberley region, Australia. Journal of Quaternary Science 29, 163–174

Response: This record is excluded from the 'high quality' record list because of an insufficient number of dates within the past 2000 years.

Rouillard, A. et al. (2016a) Evidence for extreme floods in arid subtropical northwest Australia during the Little Ice Age chronozone (CE 1400–1850). Quaternary Science Reviews 144, 107 – 122

Response: This record is excluded from the 'high quality' record list because of uncertainty in the age model, possible discontinuity in the sedimentary sequence, and an uncertain climatic control on the measured proxies. The record is suitable for qualitative comparison of climate regimes within Australasia, but does not meet the needs of the PAGES Aus2k initiative.

Rouillard, A. et al. (2016b) Interpreting vegetation change in tropical arid ecosystems from sediment molecular fossils and their stable isotope compositions: A baseline study from the Pilbara region of northwest Australia. Palaeogeography, Palaeoclimatology, Palaeoecology 459, 495–507

Response: This record is excluded from the 'high quality' record list because of uncertainty in the age model, and possible discontinuity in the sedimentary sequence. The record is suitable for qualitative comparison of climate regimes within Australasia, but does not meet the needs of the PAGES Aus2k initiative.

Kind Regards,
Alexandra Rouillard, PhD

**Low-resolution Australasian palaeoclimate records of the last 2000 years**

Bronwyn C. Dixon[1], Jonathan J. Tyler[2], Andrew M. Lorrey[3], Ian D. Goodwin[4], Joëlle Gergis[5], Russell N. Drysdale[1]

**Response to Short Comment 2**

Dear Dr. Kaufman and 2k Special Issue Data Review Team,

Thank you for your comments on 'Low-resolution Australasian palaeoclimate records of the last 2000 years'. The authors look forward to working to ensure correct data archiving is achieved.

**Essential additions for this paper:**

* 1) Expand the "Data Availability" section to include a URL to a landing page for the data compilation in this paper. The landing page should list the 22 individual datasets that were selected for this study (Table 1) plus those that were considered but not selected (Table S1).

**Response:** The following URL will become active upon publication of this manuscript:

ftp://ftp.ncdc.noaa.gov/pub/data/paleo/pages2k/dixon2017australasia/

Age control information, the original published age-depth model, the updated age-depth model, and the previously published time series for each of the 22 'high quality' data sets will be available at this URL.

Raw datasets for each of the publications in table S1 have not been collected. If the data team would prefer, it is possible to include table S1 in spreadsheet form in the online data folder.

* 2) Add Data Citations/URLs (in addition to publication citations) for each of the 22 datasets selected for this compilation. For those data not already in a persistent public repository, submit essential metadata along with the proxy time series itself and add the Data Citation/URL in Table 1.

**Response:** A column has been added to table 1, which contains data URL(s) for each of the 22 'high quality' datasets.

* 3) Include the 14C and other age control for each of the 22 records, plus the age ensembles (a primary outcome of this study), within the publicly archived files (LiPD format is recommended).

**Response:** Currently published age control information is available at the individual data URLs mentioned in the previous comment. The BACON age-depth models will be available at:

ftp://ftp.ncdc.noaa.gov/pub/data/paleo/pages2k/dixon2017australasia/

\* 4) If any of the records used in this study were also used in previous PAGES 2k databases (temperature or isotopes), please include cross references to those IDs in Table 1.

**Response:** Data IDs of records used in previous PAGES2k databases have been included in a 'Data ID' column in Table 1 in the online material.

**Strongly recommended:**

\* 5) Table S1 is a major resource for the paleoclimate community. Its value would be increased by: (a) Stating the criterion (or criteria) that was not met for each of the records that did not make the final cut. This will enable future users to easily cull the records that meet other criteria (e.g., lower resolution). (b) Naming the archive and proxy type for each record (as in Table 1). (c) Replacing the "NA" with the actual data or explain the purpose/meaning of the "NA".

**Response:** Table S1 has been updated to include the archive and proxy for each record. 'NA' was used when it was not possible to calculate the field (i.e. resolution) and/or when it was not possible to access the dataset. This will be clarified in the table caption.

\* Please also note the supplement to this comment:

http://www.clim-past-discuss.net/cp-2017-31/cp-2017-31-SC2-supplement.pdf